🔓 | **Open Peer Review** | Environmental Microbiology | Research Article

# Kinetic, genomic, and physiological analysis reveals diversity in the ecological adaptation and metabolic potential of *Brachybacterium equifaecis* sp. nov. isolated from horse feces

Adeel Farooq,[1] Myunglip Lee,[2] Saem Han,[3] Gi-Yong Jung,[4,5] So-Jeong Kim,[4] Man-Young Jung[3,6,7]

**ABSTRACT**   Brachybacterium species have been identified in various ecological niches and belong to the family *Dermabacteriaceae* within the phylum *Actinobacteria*. In this study, we isolated a novel *Brachybacterium equifaecis* JHP9 strain from horse feces and compared its kinetic, biochemical, and genomic features with those of other *Brachybacterium* strains. Moreover, comparative genomic analysis using publicly available *Brachybacterium* genomes was performed to determine the properties involved in their ecological adaptation and metabolic potential. Novel species delineation was determined phylogenetically through 16S rRNA gene similarity (up to 97.9%), average nucleotide identity (79.5–82.5%), average amino acid identity (66.7–75.8%), and *in silico* DNA-DNA hybridization (23.7–27.9) using closely related strains. This study also presents the first report of the kinetic properties of *Brachybacterium* species. Most of the *Brachybacterium* strains displayed high oxygen ($K_{m(app)}$ =1.6–24.2 µM) and glucose ($K_{m(app)}$ =0.73–1.22 µM) affinities, which may manifest niche adaptations. Various carbohydrate metabolisms under aerobic and anaerobic conditions, antibiotic resistance, mobile genetic elements, carbohydrate-active enzymes, lactic acid production, and the clustered regularly interspaced short palindromic repeats-Cas and bacteriophage exclusion systems were observed in the genotypic and/or phenotypic properties of *Brachybacterium* species, suggesting their genome flexibility, defense mechanisms, and adaptability. Our study contributes to the knowledge of the kinetic, physiological, and genomic properties of *Brachybacterium* species, including the novel JHP9 strain, which advocates for their tolerant and thriving nature in various environments, leading to their ecological adaptation.

**IMPORTANCE**   Basic physiological and genomic properties of most of the *Brachybacterium* isolates have been studied; however, the ability of this bacterium to adapt to diverse environments, which may demonstrate its role in niche differentiation, is to be identified yet. Therefore, here, we explored cellular kinetics, metabolic diversity, and ecological adaptation/defensive properties of the novel *Brachybacterium* strain through physiological and comparative genomic analysis. In addition, we presented the first report examining *Brachybacterium* kinetics, indicating that all strains of *Brachybacterium*, including the novel one, have high oxygen and glucose affinity. Furthermore, the comparative genomic analysis also revealed that the novel bacterium contains versatile genomic properties, which provide the novel bacterium with significant competitive advantages. Thus, in-depth genotypic and phenotypic analysis with kinetic properties at the species level of this genus is beneficial in clarifying its differential characteristics, conferring the ability to inhabit diverse ecological niches.

**KEYWORDS**   *Brachybacterium equifaecis* JHP9, cellular kinetics, genome distinctiveness, CRISPR-cas system, fermentation, lactic acid bacteria

Address correspondence to Man-Young Jung, myjung@jejunu.ac.kr.

Adeel Farooq, Myunglip Lee, and Saem Han contributed equally to this article. The order of authors was determined based on their significant inputs to the manuscript preparation.

The authors declare no conflict of interest.

See the funding table on p. 19.

The vast amount and diversity of bacteria on Earth, together with ever-increasing human exposure, suggest that we will continuously encounter novel bacterial isolates (1). The gut microbiota plays a vital role in the health, metabolism, and overall well-being of the host. Horses belong to a family of herbivorous mammals that possess a certain hindgut (cecum and colon) microbiota, which provide a substantial proportion of energy for horses through fermentation. Furthermore, equine gut microbiota contributes to essential physiological processes such as digestion, nutrient absorption, and immune system development (2, 3). The equine gut microbiota plays a vital role; however, available data and studies on horse microorganisms are limited. Therefore, isolating and characterizing novel microorganisms from the horse gut offer an opportunity to enhance our understanding of the microbial diversity, unique characteristics, and functional capacities associated with equine gut microbiota. Genomic versatility and the consequent physiological properties of novel strains regarding their adaptation to various ecosystems, including the horse gut, are also required.

*Brachybacterium* is a genus of gram-positive, aerobic, rod-shaped, non-spore-forming bacteria belonging to the family *Dermabacteriaceae* (4). *Dermabacteriaceae* also includes three other genera, namely, *Dermabacter*, *Devriesea*, and *Helcobacillus*. Most members of these genera isolated from diverse clinical samples are considered opportunistic human pathogens, with the exception of the genus *Brachybacterium* (5, 6). To date, *Brachybacterium* includes 24 species with validly published names ([https://lpsn.dsmz.de/genus/brachybacterium](https://lpsn.dsmz.de/genus/brachybacterium)) and has been isolated from different ecological niches (e.g., soil, plants, water, food products, and animal feces) (7). They are rarely isolated from humans, but a recent case report documented a *Brachybacterium* sp. as the causative pathogen of bloodstream infections in humans (8, 9).

Isolation of *Brachybacterium* strains from a wide range of sources demonstrates that they have adapted to diverse environmental conditions. *Brachybacterium* is a heterotroph that must cope with natural fluctuations, such as the limited availability of nutrients and oxygen, to thrive in various environments similar to other heterotrophic bacteria (10). This ability provides an evolutionary advantage that enables them to outcompete their neighbors (11). Although the basic physiological and genomic properties of most *Brachybacterium* isolates have been studied (7, 12–14), the features that contribute to niche differentiation remain unknown. The survival, diversity, and lifestyle strategies of microorganisms largely depend on their encoded ecophysiological repertoires, genomic plasticity, and cellular kinetic properties, including substrate and oxygen affinities (15). The kinetic affinity of a microorganism can be expressed using Michaelis-Menten kinetic equations, analogous to enzyme kinetics, defined by an apparent half-saturation concentration ($K_{m(app)}$) and maximal reaction rate ($V_{max}$). Based on their cellular kinetic properties, microorganisms can be explained in terms of niche differentiation as either growth rate-optimized (*r*-strategists) or growth yield-optimized (*K*-strategists) ecotypes. Thus, in-depth genotypic and phenotypic analyses of the kinetic properties of this genus at the species level will be beneficial for clarifying its differential characteristics that confer the ability to inhabit diverse ecological niches.

Consequently, this study aimed to determine the physiological, kinetic, phylogenetic, and genomic properties of *Brachybacterium* species, particularly emphasizing those of a novel *B. equifaecis* JHP9 strain isolated from horse feces. We investigated the potential adaptability of *Brachybacterium* species to natural habitats by employing comparative genomics of seven *Brachybacterium* genomes and conducting physiological analyses using reference genomes. We focused on elucidating their metabolic potential, particularly their use of various carbohydrates, adaptive strategies, and cellular kinetics. In this study, we sought to enhance our understanding of *Brachybacterium* species and their ecological significance. Specifically, by characterizing the features of strain JHP9, we aimed to provide valuable insights into its adaptive capabilities, potential applications, and the environmental role for its host, the horse.

## RESULTS

### Identification of novel strain JHP9

Phylogenetic analysis was performed based on the 16S rRNA gene sequence to identify the *B. equifaecis* JHP9 strain. Similarities of the closest relatives of this strain were identified as *B. nesterenkovii* CIP104813[T] (97.9%), *B. rhamnosum* LMG19848[T] (97.7%), *B. squillarum* THG S15-4[T] (97.5%), and *B. huguangmaarense* M1[T] (97.4%) and their phylogenetic locations reconstructed (Fig. 1A). Moreover, whole-genome phylogenetic analysis based on the core genomes of the *Brachybacterium* strains and their closely associated genomes, which was implemented using the composition vector method through the CVTree4 program, exhibited distinctive clustering of the *Brachybacterium* genomes belonging to the family *Dermabacteriaceae* based on phylogenetic distance (Fig. 1B). Among 19 families of the order *Micrococcales*, genome-based clustering revealed a close phylogenetic association between *Dermabacteriaceae* and the *Brevibacteriaceae* and *Beutenbergiaceae* families.

The genome size of strain JHP9 was 3.08 Mbp, with a 71.1% G + C content. It harbored 2,720 protein-coding sequences, 57 RNAs, and an N50 value of 754,379 bp (Table S1). Completeness of the JHP9 genome was 95%, as observed by predicting complete single-copy benchmarking universal single-copy orthologs (BUSCOs) in the genome.

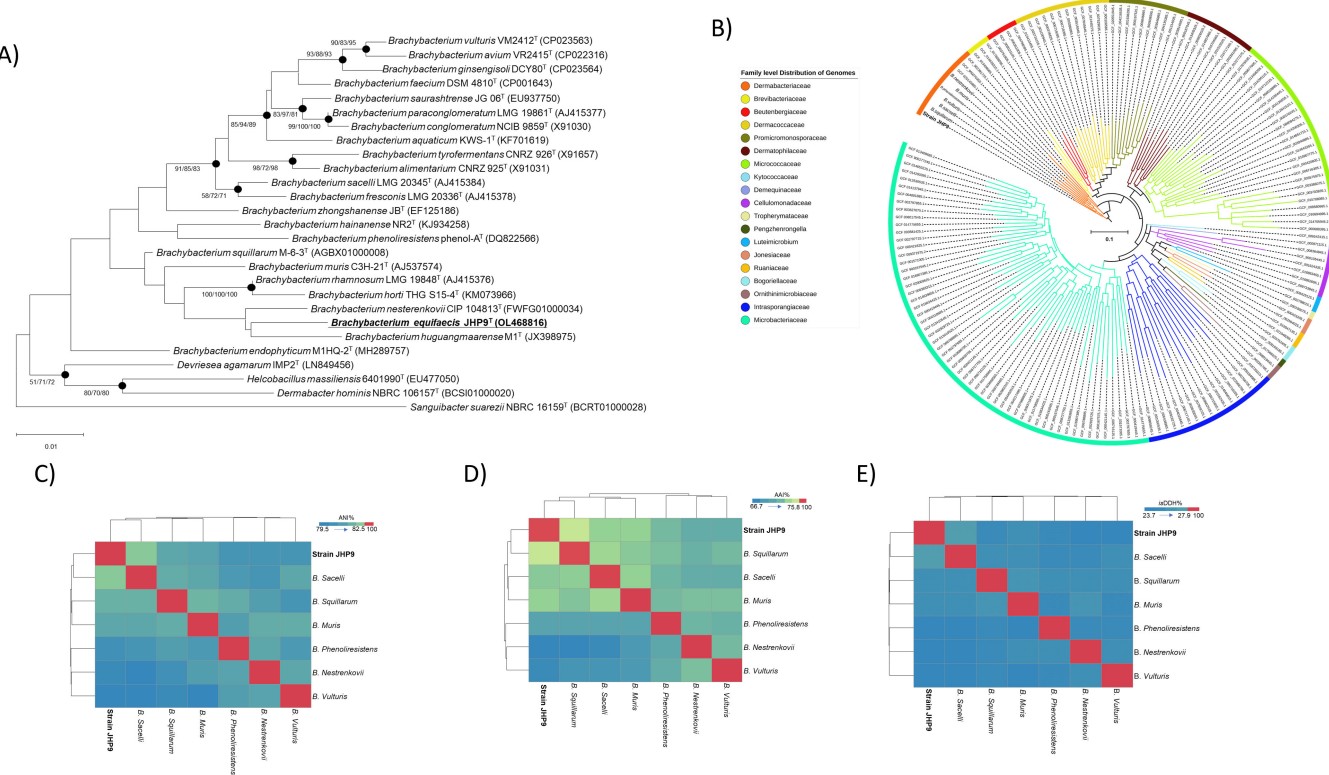

**FIG 1** Species delineation and phylogenetic clustering of *B. equifaecis* strain JHP9. (A) Maximum-likelihood tree based on the 16S rRNA gene sequences. Maximum-likelihood, neighbor-joining, and maximum parsimony algorithms with 1,000 replicates were performed for JHP9 and its closely related strains, with the number provided in parenthesis. Filled circles corresponding to nodes were also reconstructed using both neighbor-joining and maximum parsimony algorithms. *Sanguibacter suarezii* NBRC 16,159T (BCRT 01000028) was used as an outgroup. Bar indicates 0.01 substitution per nucleotide position. (B) Phylogenetic tree based on the core genome alignment of the JHP9 genome and its closely associated genomes from the phylum *Actinomycetota*. Representative genomes (163) from 19 different families of the order Micrococcales, along with *Brachybacterium* genomes (7), were downloaded from the NCBI RefSeq database, and alignment was performed using an alignment-free, composition vector-based method through the CVTree4 program. Phylogenetic clusters were visualized using iTOL v.6.5.2. The phylogenetic tree is rooted in strain JHP9, and the branch length depicts the phylogenetic distance. Similarity matrix along with the dendrograms for pairwise genome comparisons of *Brachybacterium* genomes based on (C) average nucleotide identity (ANI), (D) average amino acid identity (AAI), and (E) *in silico* DNA-DNA hybridization (*is*DDH).

These genomic characteristics are nearly in agreement with those of known *Brachybacterium* spp. (Table S1). DNA relatedness has been used as a genotypic parameter to delineate species. Structural analysis of the genome provides insights into its distinctive features. The similarity matrix calculated through comparative genome analysis showed 79.5–82.5% average nucleotide identity (ANI), 66.7–75.8% average protein identity (AAI), and 23.7–27.9 *in silico* DNA-DNA hybridization (*is*DDH) values (Fig. 1C). According to the suggested cutoff values of the ANI (<95–96%) (16, 17), AAI (<95–96%) (18, 19), and *is*DDH (70%) (18) for species delineation, the calculated values based on the comparative genome analysis results indicate that strain JHP9 is distinguished from other previously reported *Brachybacterium* species. The most closely associated strain in terms of the ANI (82.52%) and *is*DDH (27.90%) values was *B. sacelli*, whereas on the basis of the AAI value (75.82%), *B. squillarum* was the closest neighbor of strain JHP9. Genome-based phylogenetic analyses also confirmed the distinctiveness of strain JHP9 and revealed that *Brachybacterium* genomes are closely associated with members of *Brevibacteriaceae*, a family of the phylum *Actinomycetota*, as shown in Fig. 1. Taken together, based on physiological attributes, phylogenetic analysis of the 16S rRNA gene, genome-based phylogenetic analyses, as well as the ANI, AAI, and *is*DDH values, strain JHP9 can be proposed as a representative novel species of the genus *Brachybacterium* with the name *B. equifaecis* sp. nov.

## Morphological, physiological, and biochemical characterization

Cells of the novel JHP9 strain were gram positive, non-motile, and ellipse shaped, having a diameter length of $0.8 \times 1.8$ μm, as shown via transmission electron microscopy (TEM) (Fig. S1). Structural observations confirmed the rod-shaped morphology of strain JHP9 during its exponential growth phase, whereas it was coccoid during the stationary phase. This behavior was similar to that of *B. nesterenkovii* (20). Colonies were cream colored, smooth, and glistening and grew well on tryptone soya agar (TSA) incubated at 30℃ for 7 d. Growth of strain JHP9 occurred at 18–37 ℃ (optimum, 30℃), pH 6.0–9.0 (optimum, pH 7.0), and 0–10% (wt/vol) NaCl (optimum, 0.5–1%) (Table 1). The strain hydrolyzed starch and DNA but not casein, cellulose, Tween 20, and Tween 80. The hydrolytic activity of *β*-glucosidase was positive but negative for protease, which was the same for all the tested *Brachybacterium* species (Table S2). Positive reactions were detected in the catalase but not in the oxidase tests. Strain JHP9 was negative for indole formation, the dissimilatory reduction of nitrate to nitrite and dinitrogen, and urease activity. The unique enzymatic activity of strain JHP9 against alkaline phosphatase and *β*-glucuronidase was identified (Table S2). All three tested *Brachybacterium* species were positive for the assimilation of D-glucose, *N*-acetyl-D-glucosamine, and gluconate but negative for L-arabinose and caprate (Table S2). The following substrates were used for fermentation: D-ribose, D-galactose, D-glucose, D-mannose, methyl α-D-glucopyranoside, esculin, D-cellobiose, D-lactose, D-trehalose, D-raffinose, and glycogen (Table S2). Among these substrates, D-ribose, D-galactose, D-glucose, D-mannose, and esculin could also be used as universal fermentation sources for other *Brachybacterium* strains, but *N*-acetyl-D-glucosamine was unique only to strain JHP9 (Table S2).

Two-dimensional thin-layer chromatography (TLC) showed that the major polar lipids in strain JHP9 were phosphatidylglycerol (PG), diphosphatidylglycerol (DPG), and glycerolipids (GL) (Fig. S2). This profile was similar to that of other *Brachybacterium* species (20, 34). Its cell wall peptidoglycan contained *meso*-diaminopimelic acid (DAP), alanine, and glutamic acid as the major amino acids, whereas MK-7 (99.48%) and MK-8 (0.52%) were the predominant menaquinones (Table 1), and $C_{15:0}$ anteiso, $C_{19:0}$ cyclo *ω8c*, $C_{17:0}$ anteiso, and $C_{16:0}$ iso were the main fatty acid (>5%) components (Table S3). All three tested *Brachybacterium* species shared $C_{15:0}$ anteiso as the most abundant branched-chain fatty acid. Based on chemotaxonomic analysis, strain JHP9 was proposed to belong to the genus *Brachybacterium*. Carbohydrate fermentation was also observed in the strain, although it was weaker than that observed in *B. nesterenkovii* JCM 11648[T]. Based on the susceptibility test results, all three *Brachybacterium* species were

**TABLE 1** Phenotypic features of the strain JHP9 and its closely associated *Brachybacterium* species[f]

| Characteristic | 1 | 2 [a,c] | 3 [b] |
|---|---|---|---|
| Isolated site | Horse feces | Milk product | Lake sediment |
| Growth in medium | TSB or MRS | TSB + yeast extract[e] | TSA + yeast extract[e] |
| Growth in temperature (°C) | | | |
| Growth range (optimum) | 18–37 (21) | 15–42 (21–33) | 18–40 (21, 25, 26) |
| Growth NaCl (wt/vol) | | | |
| Growth range (optimum) | 0–10 (0.5–1) | 0–1 (0–0.5) | 0–15 (ND[d]) |
| Growth pH | | | |
| Growth range (optimum) | 6–9 (7) | 6–10 (6–9) | 4–8 (7, 8) |
| Oxidase | − | + | − |
| Catalase | + | + | + |
| Major menaquinone | MK7 | MK7 | MK7 |
| Peptidoglycan | Meso-DAP, Ala, Glu | Meso-DAP, Ala, Gly, Asp, Glu | Meso-DAP, Ala, Gly, Asp, Glu |
| Polar lipid profile | DPG, PG, GL | DPG, PG, GL | DPG, PG, GL, PL |
| DNA G + C content (mol%) | 71 | 70 | 71 |
| Macromolecule degradation | Starch, DNA | Starch | ND |

[a]Data from (20).
[b]Data from (34).
[c]Data from (4).
[d]ND, no data available.
[e]0.3% yeast extract in media.
[f]1, *Brachybacterium equifaecis* JHP9T; 2, *Brachybacterium nesterenkovii* JCM11648T; 3, *Brachybacterium huguangmaarense* JCM30544T. meso-DAP, meso-diaminopimelic acid; DPG, diphosphatidylglycerol; PG, phosphatidylglycerol; GL, glycolipid; PL, phospholipid; +, positive; −, negative.

susceptible to ampicillin, gentamicin, streptomycin, and tetracycline. However, these strains were resistant to kanamycin. Additionally, strain JHP9 displayed resistance to chloramphenicol, whereas the remaining strains demonstrated susceptibility to it (Table S4).

## Comparative genomics of *Brachybacterium*

A pangenomic analysis of seven *Brachybacterium* genomes, including that of strain JHP9, showed that these organisms share 1,041 core genes. The pangenome comprised 21,264 genes shared among *B. muris* ($n = 2,736$), *B. nesterenkovii* ($n = 2,455$), *B. phenoliresistens* ($n = 3,509$), *B. equifaecis* JHP9 ($n = 2,660$), *B. sacelli* ($n = 3,902$), *B. squillarum* ($n = 2,679$), and *B. vulturis* ($n = 3,115$) (Fig. 2A and B). Functional assignment of the pangenome revealed variations in the distribution of genes within the different metabolic fractions (Fig. 2C). The core genome exhibited a prominent presence in essential processes, such as amino acid metabolism, translation, nucleotide metabolism, replication and repair, energy metabolism, folding, sorting and degradation, cell growth and death, and transcription. In contrast, the accessory genome showed significant representation of functions related to carbohydrate metabolism, cellular community, terpenoid and polyketide metabolism, drug resistance, and xenobiotic biodegradation and metabolism. In contrast, the unique genome demonstrated substantial representation of functions associated with membrane transport, xenobiotic biodegradation and metabolism, lipid metabolism, biosynthesis of secondary metabolites, glycan biosynthesis and metabolism, cell motility and transport, and catabolism (Fig. 2C). These core, accessory, and unique *Brachybacterium* genomes may support their heterotrophic lifestyle, niche differentiation, and adaptation features.

Thus, the metabolic potential of the novel JHP9 strain was further explored to uncover the complete or nearly complete metabolic pathways encoded by its genome. In-depth analysis revealed that its ABC transportation system consisted of various components, including oligosaccharides (nineeach), phosphates (four), minerals (two), metallic cations (two), and monosaccharides (one) (Fig. 3). Furthermore, metabolic pathways for carbohydrates, such as maltose, cellobiose, starch, galactose, D-mannitol, trehalose, and mannose, were identified within the genome. Additionally, it exhibited the

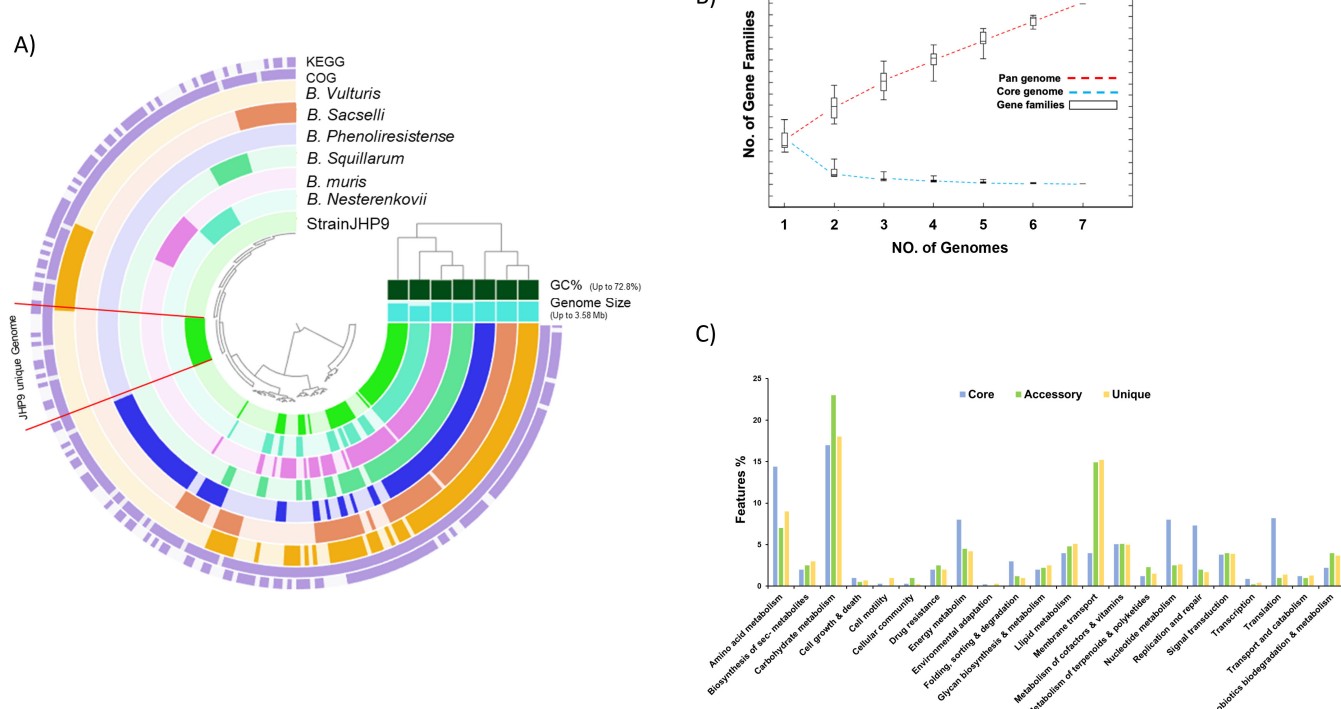

**FIG 2** Pangenome analysis of *Brachybacterium equifaecis* strain JHP9. (A) Each circle represents a genome, and each radius, a gene family. Core genome families are localized on the right, whereas some of these core families have more than one homologous gene per genome. The shell genome is observed in the middle and at the bottom of the figure, whereas dispensable genome and singletons are represented on the left. Deep ring colors represent the presence of the respective gene clusters, whereas faded ring colors indicate their absence. (B) A typical pan/core plot comprised of seven *Brachybacterium* genomes. The core genome decreases when more genomes are added, whereas the pangenome increases upon the addition of genomes. (C) Kyoto Encyclopedia of Genes and Genomes (KEGG) distribution of core/accessory/unique genomes in the *Brachybacterium* strains.

presence of the pentose phosphate pathway, terpenoid backbone biosynthesis, pyruvate metabolism, amino acid metabolism, GL metabolism, nucleotide sugar biosynthesis, and the metabolism of cofactors and vitamins. Genomic analysis of this bacterial strain revealed the presence of pathways related to oxidative phosphorylation, demonstrating its capacity for efficient energy production through aerobic respiration (Fig. 3). Notably, similar metabolic potential patterns were identified in other *Brachybacterium* genomes, except for variations in the presence of ABC transporters (Fig. S6).

Detailed genomic analyses revealed genes coding for antibiotic resistance, the clustered regularly interspaced short palindromic repeats (CRISPR) system, mobile genetic elements (MGEs), carbohydrate-active enzymes (CAZymes), and stress responses (Table 2; Fig. S4; Supplementary File 2). The JHP9 genome was found to carry three antibiotic resistance genes (ARGs) that confer resistance to two distinct groups of antibiotics. These ARGs were *gyrA* (locus number; Bequi_11645) and *gyrB* (Bequi_11640), encoding resistance against quinolones, while *blaIII* (Bequi_03615) gene determining resistance against beta-lactam antibiotics. In contrast, none of the other *Brachybacterium* genomes carried any of the known resistance determinants. Moreover, no virulence factors and prophages were detected in the *Brachybacterium* genomes. The Type I-E CRISPR-Cas system was observed in the JHP9, *B. nesterenkovii*, *B. squillarum*, and *B. vulturis* genomes. The CRISPR-Cas locus comprises three arrays and eight associated proteins, including *cas2e*, *cas1e*, *cas6e*, *cas5e*, *cas7e*, *casB*, *casA*, and *cas3* (Bequi_14035-14070), encompassing the type I-E CRISPR system. The *B. nesterenkovii*, *B. squillarum*, and *B. vulturis* strains also expressed these eight associated proteins; *B. nesterenkovii* had two CRISPR arrays, and *B. squillarum* and *B. vulturis* each had one array, as described in Table 2; Supplementary File 2. Moreover, the bacteriophage exclusion (BREX) system

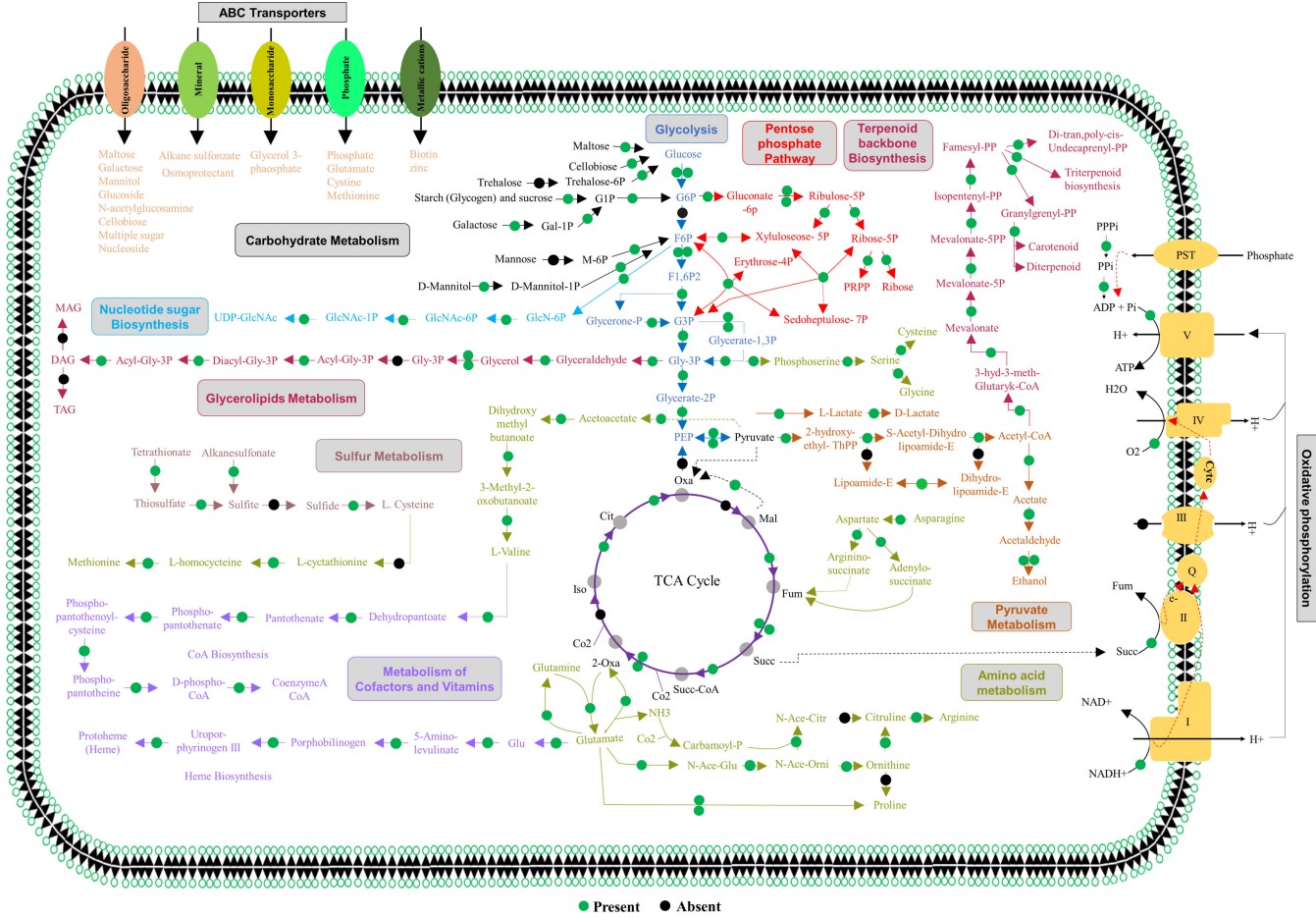

**FIG 3** Schematic metabolic potential of the novel *B. equifaecis* strain JHP9. Metabolism pathways were reconstructed based on KEGG annotation. Black dots refer to the absence of respective proteins. The genes that could be detected in strain JHP9 are shown with the green-colored dots. This figure illustrates the genomic potential of strain JHP9 in the uptake of diverse carbohydrates, minerals, and metal ions. It also highlights the involvement of specific proteins in the ABC transportation system for oligosaccharides, monosaccharides, minerals, phosphates, and metallic cations. Additionally, the figure showcases the enzymes associated with the utilization of various carbohydrates, such as galactose, glucose, trehalose, glycogen, lactose, and cellobiose metabolism, as well as pathways such as the pentose phosphate pathway, pyruvate metabolism, amino acid metabolism (arginine, ornithine, proline, and asparagine), cofactors and vitamins (CoA and heme biosynthesis), terpenoid biosynthesis, glycerolipid metabolism, and oxidative phosphorylation.

(Bequi_11770-11790) was identified in the JHP9 genome. Several MGEs were found in *Brachybacterium* genomes, including IS30, ISL3, IS256, and IS110, which were distributed over seven, six, six, and six genomes, respectively (Table 2; Supplementary File 2).

Based on CAZyme analysis, *Brachybacterium* genomes (*n* = 7) contained a number of carbohydrate-active genes coding for glycosyl transferases (GTs) (21–32), glycoside hydrolases (GHs) (11–20, 34), and carbohydrate esterases (CEs) (0–4), indicating their ability to utilize a variety of carbohydrates (Table 2; Supplementary File 2). Similar to most of the *Brachybacterium* strains, strain JHP9 has canonical defense systems that aerobic organisms need to survive against oxidative stress, including superoxide dismutase, catalase, thioredoxin, and glutathione peroxidase (Supplementary File 2). Moreover, most *Brachybacterium* genomes contained genes related to oxidative and osmotic stress, such as cold shock proteins and choline (BetT) biosynthesis (Table 2 and Supplementary File 2).

Furthermore, genome annotation revealed that JHP9 and other *Brachybacterium* genomes encode genes for two types of terminal oxidases, including cytochrome *d* ubiquinol subunits I and II (CydAB), which predominate under low-aeration growth conditions with high oxygen affinity. Cytochrome *c* oxidase subunits I, II, and IV (COX1,

**TABLE 2** Comparative genomic analyses of CAZymes, stress-related genes, CRISPR systems, MGEs, and ARGs encoded by the *Brachybacterium* genomes (*n* = 07)Supplementary file 2[b]

| Genomes | CAZymes | Stress-related genes | CRISPR system | | MGEs | ARGs |
|---|---|---|---|---|---|---|
| | | | CRISPR arrays | Associated proteins | (family) | |
| Strain JHP9 | GT (27) | SOD (1) | 3 | 8 | IS3 (07) | Quinolones |
| | GH (11) | GPx (2) | | | IS30 (06) | (2) |
| | CE (4) | Trx (2) | | | IS481 (06) | Beta-lactam (1) |
| | | CAT (1) | | | | |
| | | Csp (2) | | | | |
| | | BetT (1) | | | | |
| *B. muris* | GT (30) | SOD (1) | ND[a] | ND | IS21 (135) | ND |
| | GH (14) | Trx (2) | | | IS3 (18) | |
| | CE (1) | CAT (1) | | | IS1380 (14) | |
| | | Csp (3) | | | | |
| | | BetT (1) | | | | |
| | | EctC (1) | | | | |
| *B. nesterenkovii* | GT (30) | SOD (1) | 2 | 8 | IS110 (9) | ND |
| | GH (15) | GPx (1) | | | ISL3 (8) | |
| | CE (2) | Trx (2) | | | IS256 (5) | |
| | | CAT (1) | | | | |
| | | Csp (2) | | | | |
| | | BetT (1) | | | | |
| | | EctC (1) | | | | |
| *B. phenoliresistens* | GT (29) | SOD (2) | ND | ND | IS110 (03) | ND |
| | GH (30) | GPx (2) | | | IS30 (1) | |
| | CE (1) | Trx (1) | | | | |
| | | CAT (2) | | | | |
| | | Csp (3) | | | | |
| | | BetT (1) | | | | |
| | | EctC (1) | | | | |
| *B. sacelli* | GT (25) | SOD (3) | ND | ND | IS3 (20) | ND |
| | GH (34) | GPx (2) | | | IS481 (11) | |
| | CE (0) | Trx (1) | | | IS256 (06) | |
| | | CAT (2) | | | | |
| | | Csp (3) | | | | |
| | | BetT (1) | | | | |
| | | EctC (2) | | | | |
| *B. squillarum* | GT (32) | SOD (1) | 1 | 8 | IS30 (7) | ND |
| | GH (13) | GPx (2) | | | IS481 (6) | |
| | CE (2) | Trx (1) | | | IS110 (5) | |
| | | CAT (1) | | | | |
| | | Csp (3) | | | | |
| *B. vulturis* | GT (22) | SOD (2) | 1 | 8 | IS110 (6) | ND |
| | GH (13) | GPx (1) | | | IS1380 (5) | |
| | CE (1) | Trx (1) | | | IS30 (2) | |
| | | CAT (1) | | | | |
| | | Csp (3) | | | | |
| | | BetT (2) | | | | |
| | | EctC (2) | | | | |

[a]ND, not detected.
[b]The type and number of attributes are given here, while a detailed description is given in Supplementary file 2. GT, glycosyl transferases; GH, glycoside hydrolases; CE, carbohydrate esterases; SOD, superoxidase dismutase; GPx, glutathione peroxidase; Trx, thioredoxin; CAT, catalase; CspA, cold shock protein; BetT, choline transporter; EctC, ectoine synthase; T3PKS, 5-acetyl-5,10-dihydrophenazine-1-carboxylic acid.

2, and 4) are heme-copper oxidases (HCO) that prefer high-aeration conditions with low oxygen affinity (35). The *cydA* and *cydB* genes in the cytochrome *d* ubiquinol

cluster encode two polypeptide subunits of the cytochrome *d* terminal oxidase complex, whereas *cydC* and *cydD* are involved in the assembly of cytochrome *bd*-I, a terminal oxidase of the respiratory chain required for growth under low oxygen conditions. A cluster of these four genes, which constitutes cytochrome *d* ubiquinol, was encoded by all *Brachybacterium* genomes (Fig. S5). Moreover, a COX cluster comprising the *cox1*, *cox2*, and *cox4* genes was present in all *Brachybacterium* genomes, as shown in Fig. S5.

## Lactic acid production by *Brachybacterium*

Quantitative analysis of lactic acid production during fermentation by strain JHP9 and its closely associated strains was performed using six different sugar compounds. Our results revealed a significant difference in lactic acid production between the *Brachybacterium* strains, and strain JHP9 produced a higher concentration of lactic acid from mannose and sucrose than that of other *Brachybacterium* strains (Fig. 4). In contrast, *B. nesterenkovii* JCM11648[T] produced the highest lactic acid concentration from galactose, lactose, and glucose. The *B. huguangmaarense* JCM 30544[T] strain only produced a significant concentration of lactic acid from mannose, glucose, and sucrose. The results of lactic acid production were in accordance with those of the fermentation activity (Fig. 4; Table S2), except for some sugars, such as galactose, which was positive and did not produce lactic acid. Notably, under our experimental conditions, the lactic

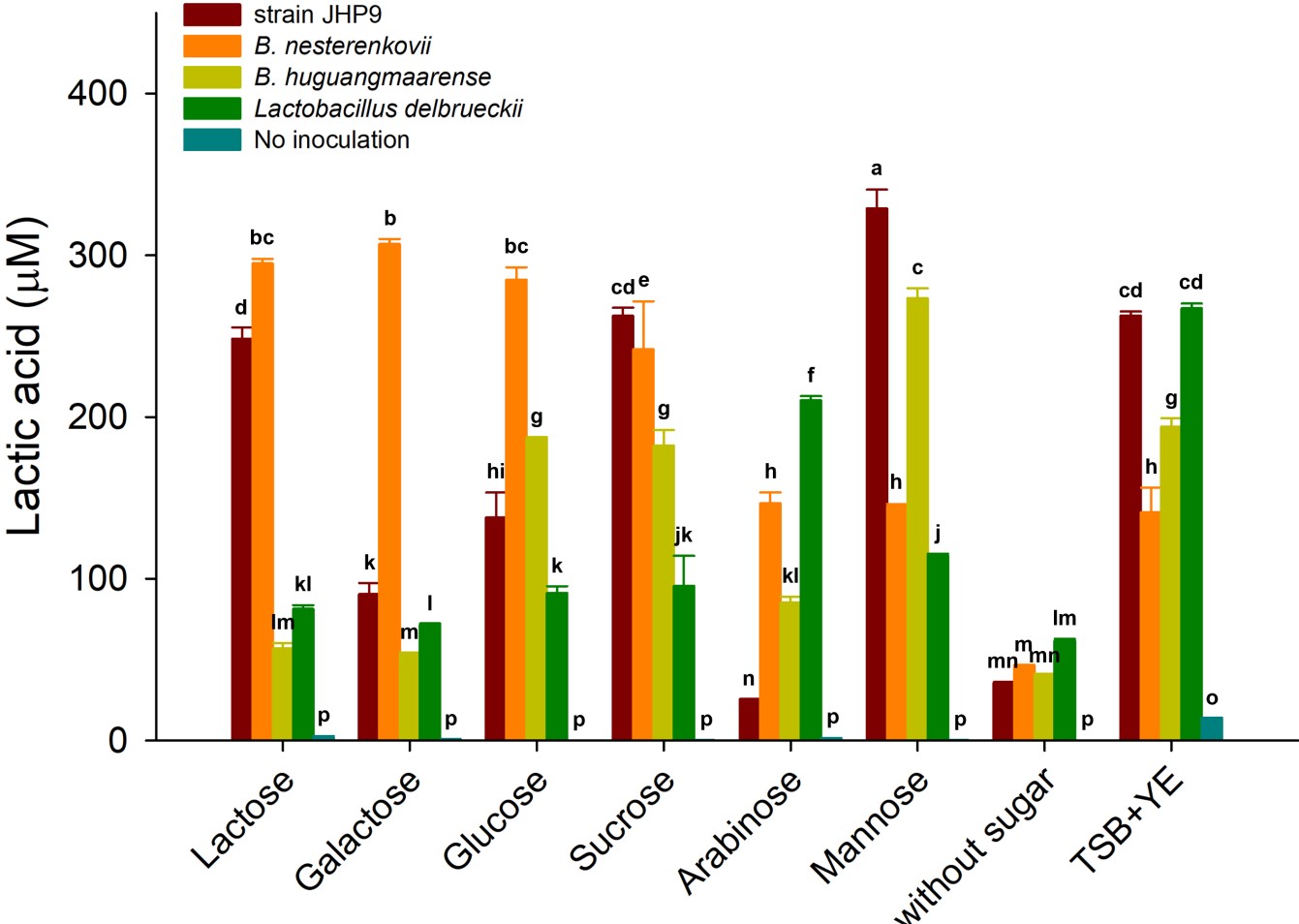

**FIG 4** Lactic acid production of three *Brachybacterium* strains. A lactic acid bacterial strain, *Lactobacillus delbrueckii* subsp. *bulgaricus* KCTC 3769[T], was also tested for comparison. Artificial freshwater medium without sugar and no inoculum served as a negative control. Error bars represent the standard deviation for $n \geq 3$ biological replicates. Significant differences between treatments in each strain are indicated by different letters (one-way analysis of variance, Tukey's test, $P < 0.05$).

acid-producing bacterium (LAB), *Lactobacillus delbrueckii* subsp. *bulgaricus* KCTC 3769[T], used as a positive control strain, did not produce higher levels of lactic acid, except when arabinose was used as a substrate (~210 µM).

## Substrate and oxygen affinity in *Brachybacterium*

Cellular respiration kinetics for glucose and oxygen affinity were determined using strain JHP9 and other closely associated *Brachybacterium* species by measuring glucose- and oxygen-dependent oxygen consumption in microrespirometry (MR) experiments (Fig. S6). The stoichiometry of glucose ($C_6H_{12}O_6$) and $O_2$ consumption was always near 1:6 (mean = 1:6.11; SD = 0.12; $n$ = 15), which was expected for heterotrophs that could use glucose as a substrate. The apparent $K_{m(app)}$ for glucose of strain JHP9 was determined to be 1.9 ± 0.3 µM, which was the highest affinity calculated among all tested *Brachybacterium* strains. The other tested *Brachybacterium* strains, excluding *B. horti* ($K_{m(app)}$ =24.26 µM), had a similar affinity range for glucose ($K_{m(app)}$ =2.04–10.6 µM). However, the cellular kinetic affinity for oxygen of all tested *Brachybacterium* strains was in a similar range ($K_{m(app)}$ =0.73–1.22 µM) (Fig. 5). The oxygen uptake affinity was much higher than that of heterotrophic bacteria (*Escherichia coli* and *Pseudomonas chlororaphis*) (22, 23, 36, 37) and autotrophic ammonia oxidizers (ammonia-oxidizing bacteria and archaea) (24) but lower than that of the LAB strain, *Bifidobacterium bifidum* (25). The affinity for glucose and oxygen in *Brachybacterium* strains was lower than that of the marine bacterial consortium in oligotrophic habitats.

## DISCUSSION

In this study, we described the phenotypic and genotypic characteristics of a novel *B. equifaecis* JHP9 strain, along with those of closely related *Brachybacterium* strains, using physiological and genomic analyses. Interestingly, *Brachybacterium* species possess

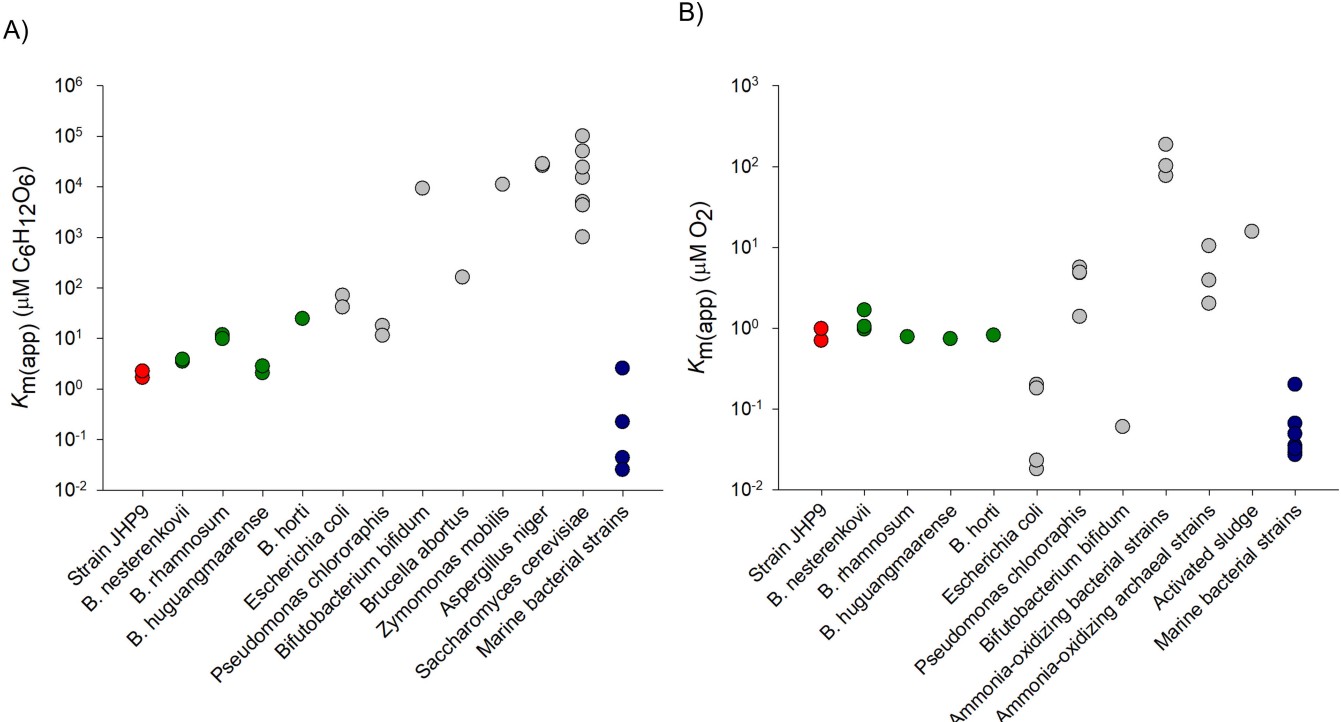

**FIG 5** Comparison of the whole-cell apparent half-saturation constants ($K_{m(app)}$) for glucose and oxygen between *Brachybacterium* strains and other microbes. (A) Glucose and (B) oxygen uptake affinities are shown with distinctive colors. The $K_{m(app)}$ values measured in this study are highlighted in red for strain JHP9 and green for other tested *Brachybacterium* strains. The other $K_{m(app)}$ values were retrieved from previous studies (22–24, 36–41). The individual Michaelis-Menten plots for each strain determined in this study are presented in Fig. S6.

properties that facilitate their versatile lifestyles and ecological adaptations. These characteristics encompass their ability to survive under various temperature, pH, and salt concentration conditions, their ability to utilize a variety of carbohydrates, and their high affinity for oxygen and glucose. Collectively, these traits facilitate successful adaptation and survival in specific environmental settings. Furthermore, analysis of the novel JHP9 strain revealed unique factors contributing to its defense mechanism, including antibiotic resistance, relatively high affinity for oxygen and glucose, and efficient utilization of sugars. These traits reflect the adaptation of strain JHP9 to the niche environment of the horse intestine.

According to the competitive exclusion principle of ecology, two species competing for the same limiting resource cannot coexist under stable environmental conditions (26). However, the co-occurrence of microbes competing for the same substrate has been verified in various natural and engineered environments. This observation challenges the traditional notion of competitive exclusion and prompts inquiries into the physiological traits that enable niche adaptation within microbial communities. For heterotrophic bacteria, such as *Brachybacterium*, efficient utilization of substrates (carbohydrates) requires the metabolic potential to obtain energy from available resources (21).

Carbohydrate utilization involves the hydrolysis of complex polysaccharides, which requires multiple CAZymes owing to the structural diversity of carbohydrates (27). *Brachybacterium* genomes contain genes encoding GTs, CEs, and GHs that hydrolyze cellular carbohydrates (28). The prevalence of these enzymes enables *Brachybacterium* strains, including JHP9, to efficiently utilize diverse carbohydrates as a competitive advantage against bacterial species using the same substrate (29). Although we did not experimentally validate specific functions of the predicted CAZymes, their ability to produce lactic acid from five different sugars (Fig. 4), assimilate five different carbon sources, and ferment 11 distinctive carbohydrates (Table S2) implies their potential involvement in carbohydrate utilization. Nevertheless, it is crucial to acknowledge that the roles of CAZymes in sugar fermentation can differ depending on the bacterium and its metabolic capabilities.

Functional annotation of the genomes revealed the presence of proteins involved in carbohydrate metabolism, including glycolysis and the tricarboxylic acid cycle, which are crucial components of the central carbon dissimilation pathway. *Brachybacterium* genomes encoded nearly all the genes required for various metabolic pathways, including carbohydrate metabolism, membrane transport (ABC transport system), and amino acid metabolism (Fig. 3; Fig. S3). Specifically, the unique *Brachybacterium* genome contained ABC transporter systems that are responsible for transporting oligosaccharides, monosaccharides, minerals, phosphates, and metallic cations (as shown in Fig. 3). This allows *Brachybacterium* spp. to cope with the chemical constraints of their natural habitats, leading to niche adaptations. Additionally, the accessory genome is involved in carbohydrate metabolism (Fig. 2C), suggesting its potential contribution to niche differentiation. Furthermore, the core *Brachybacterium* genome was involved in amino acid metabolism, suggesting that it supports a heterotrophic lifestyle. The utilization of external sources of organic compounds, including amino acids, is vital for their growth and energy production.

Lactic acid production via microbial fermentation offers advantages over chemical synthesis, including simplicity of operation, low substrate costs, moderate pressure and temperature conditions, reduced risk of contamination, low energy consumption, and improved environmental performance (30). Various microbial species, including bacteria, fungi, yeast, cyanobacteria, and algae, produce lactic acid using carbohydrates as the only or primary carbon source (31). LAB strains are predominantly anaerobic and utilize pyruvic acid, an intermediate in the Embden-Meyerhof pathway, as a substrate for lactate production. The conversion of pyruvic acid to lactate is facilitated by the activity of L- or D-lactate dehydrogenase (LDH) enzymes. Notably, *Brachybacterium* genomes also encode enzymes involved in lactate fermentation, which is consistent

with the phenotypically expressed fermentation abilities of these strains, as shown in Fig. 4. The genome of strain JHP9 encodes L-LDH (*ldh*; Bequi_03000), D-LDH (*ldhd*; Bequi_04310), and lactate racemase (*larA-C*; Bequi_13030) on distinct contigs, mirroring their presence in other tested *Brachybacterium* strains. This genetic evidence suggests that these enzymes may play a crucial role in producing L- and D-lactic acid through the fermentation processes employed by *Brachybacterium* strains. Under our experimental conditions, the *Brachybacterium* strains produced higher levels of lactic acid than that of the *Lactobacillus* LAB strain. Under various growth conditions, *Lactobacillus* strains may behave as heterofermenters and produce not only lactic acid but also acetate, formate, ethanol, diacetyl, acetoin, and $CO_2$ (32). Heterofermentative LAB can use the phosphogluconate and phosphoketolase pathways to metabolize hexose and pentose sugars, respectively. Previously, it was observed that lactic acid production by LAB through glucose consumption was lowest in MRS (De Man–Rogosa–Sharpe) media; instead, high acetate generation was observed (33), which might be due to fluctuating sugar and protein contents of the media and the incubation time. Similarly, the product of D-galactose fermentation in *B. huguangmaarense* was not lactic acid (Fig. 4; Table S2).

Although all the tested *Brachybacterium* strains demonstrated kanamycin resistance, strain JHP9 exhibited phenotypic resistance to chloramphenicol and kanamycin. Interestingly, our analysis revealed quinolone- and ampicillin-resistance determinants in the JHP9 genome, indicating its potential resistance to these antibiotics. However, we could not identify any known kanamycin or chloramphenicol resistance genes in the JHP9 genome. The observed phenotypic resistance may be attributed to mutated genes that are not homologous to the known resistance determinants for kanamycin and chloramphenicol. Alternatively, this could be due to the expression of cryptic genes that were not evaluated in this study. Conversely, the JHP9 genome carried beta-lactam-resistance genes despite exhibiting susceptibility to the corresponding antibiotic when tested phenotypically. The lack of phenotypic resistance associated with the beta-lactam-resistance gene, *blaIII*, may be attributed to several factors. These factors include mutations or genetic variations within the gene or its regulatory elements, absence of specific inducers or repressors, and genetic regulatory mechanisms (42–44). Although strain JHP9 harbors quinolone-resistance genes, its phenotypic resistance to antibiotics belonging to this class was not evaluated in this study (Table S4). *Brachybacterium* strains were reportedly resistant to ampicillin, cefazolin, vancomycin, erythromycin, rifampicin, gentamicin, clindamycin, and tetracycline (34, 45). However, this is the first report of chloramphenicol resistance and the identification of quinolone resistance determinants in any *Brachybacterium* species. Resistance to antibiotics provides an advantage for the propagation of strain JHP9, where it can outcompete susceptible isolates in a stressed environment, such as under an antibiotic selective pressure. The possible acquisition of chloramphenicol and quinolone resistance from the host (horse gut) indicates the excessive use of antibiotics to treat infections and for prophylactic purposes in livestock farming (46). Strain JHP9 may spread the antibiotic resistance trait by transferring the encoded ARGs to other closely associated human pathogenic bacteria via horizontal gene transfer.

*Brachybacterium* genomes also contained defense- and stress-related genes, as well as CRISPR-Cas systems, specifically the type I-E CRISPR system, with the JHP9 genome additionally possessing the BREX system. These systems enhance defense and immunity against foreign DNA and lytic and temperate phages (47, 48). Moreover, their genomes contained oxidative and osmotic stress-related genes (Table 2), which could enhance their ability to propagate in various ecosystems. Additionally, the presence of MGEs, including IS3, IS30, and IS481, in the JHP9 genome (Table 2), indicates its genetic variability, which consequently leads to adaptive evolution. Transposable elements are agents of genetic variability in bacteria because they are a source of adaptive evolution through genome diversification (49). In addition, MGEs are thought to enhance the defense mechanisms of bacteria (50). MGEs are known to transpose associated genes

within the genome, which not only produces genome diversity, but also enhances the phenotypic expression of colocalized genes (51).

This study is the first to report the glucose and oxygen affinities of any *Brachybacterium* species, including those within the *Dermabacteriaceae* family. All tested strains, including strain JHP9, had a relatively high affinity (with a low $K_{m(app)}$ value) for glucose and are comparable with those of *E. coli* (38) and *P. chlororaphis* (23) (see Fig. 5). These traits confer essential competitive advantages to heterotrophic bacteria (39, 40) and likely provide similar benefits to *Brachybacterium* species under oligotrophic conditions, such as in marine bacterial populations with 1–2 orders of magnitude higher glucose affinity (40). Therefore, whether *Brachybacterium* strains can persist at extremely low *in situ* substrate concentrations remains to be tested. For example, highly different substrate affinities have been observed for *E. coli* when grown under oligotrophic versus copiotrophic conditions (41). Therefore, a high glucose affinity might endow various fitness strategies to JHP9 and other *Brachybacterium* strains.

The final step of aerobic respiration involves a terminal oxidase, a membrane-associated protein that reduces $O_2$ to $H_2O$. Genomic analysis of strain JHP9 revealed the presence of genes encoding terminal oxidases, including two different terminal oxidases (cytochrome *bd* oxidase and HCO), as shown in Fig. S5, which are considered high- and low-oxygen affinity oxidases (36). Cytochrome *bd* oxidases play crucial physiological roles in enabling bacterial survival and reproduction under adverse environmental conditions. Thus, the prevalence of these terminal oxidases in the JHP9 genome not only provides it with a high oxygen affinity, which enhances its adaptability to various niches with diverse living conditions (52) but also enables its survival under toxic conditions. As described above, high- and low-affinity terminal oxidases are classified based on their different $O_2$ affinities (36, 53). Strain JHP9 has both high- and low-affinity terminal oxidase genes, similar to *E. coli* and marine bacterial strains, and the high-affinity terminal oxidases may enable these bacteria to maintain high levels of respiratory activity even when the $O_2$ concentration decreases. In addition, the oxygen affinity of washed cell suspensions of an LAB strain, *Bifidobacterium*, was high ($K_{m(app)}$ =0.06 µM) (25) and likely resulted from the function of a high-affinity cytochrome *d* oxidase (54). Even under high $O_2$ conditions, *E. coli* expressed high-affinity terminal oxidases and were incubated under fully aerobic conditions; thus, they may exhibit relatively low $K_{m(app)}$ values similar to those of *Brachybacterium* strains. It was suggested that the $K_{m(app)}$ value changes were correlated with the availability of an electron donor (55) and those for glucose oxidation affected by the oxygen concentration in *P. chlororaphis* culture; therefore, the oxygen uptake kinetics could be related to substrate affinity (23). It has also been shown that the different cell sizes (with different surface area-to-volume ratios) of various ammonia-oxidizing autotrophs are correlated with their kinetic affinity (15) and show different $K_{m(app)}$ values for different cell sizes of a single marine bacterial strain. Therefore, the microbial aerobic respiration rate is affected by various factors, including cell size, physiological growth conditions, nutrient availability, and $O_2$ concentration, which eventually control oxygen and substrate affinity. Our results imply that *Brachybacterium,* including strain JHP9, has a high affinity for glucose and oxygen. The *K-strategist* microbes grow slowly but can compete at low substrate concentrations because of their high substrate affinity (39). Therefore, members of the family *Dermabacteriaceae*, including strain JHP9, tend to utilize glucose and oxygen to survive in temperate ecosystems; hence, such a high substrate affinity is advantageous to allow them to adapt and colonize the host while competing with specialists in the same niche (56). Furthermore, the high GC content observed in *Brachybacterium* genomes (Table 1) contributes to the stability of their DNA (44). This characteristic implies that they have the capacity to flourish within specific temperature ranges and adapt to their ecological niches.

The metabolic potential, defense strategies, and physiological characteristics of *Brachybacterium* spp. contribute to their niche adaptation to specific environments, including the unique adaptation of strain JHP9 to horse intestines. High resource diversification can lead to gradual ecological specialization, in which bacterial

populations progress toward an optimal phenotype. However, resource or host generalism, which confers ecological advantages, has also been widely observed in nature and confers an ecological advantage (57). The adaptability of *Brachybacterium* strains may be attributed to the development, industrialization, and globalization of livestock farming that has created open niches in which bacteria have expanded their hosts. Livestock husbandry and habitation facilitate close contact between different species, thereby providing opportunities for bacterial transmission from one host to another (58). This transmission drives the acquisition of a broad range of traits that are necessary for environmental adaptation. Strain JHP9 exhibits features such as carbohydrate utilization, defense mechanisms, and affinity for oxygen and glucose, which contribute to its successful adaptation and survival in the horse gut environment.

In this study, we comparatively characterized various *Brachybacterium* species, with a particular focus on *B. equifaecis* JHP9, to assess their ecological adaptation and metabolic potential. However, the physiological properties deduced from *in vitro* and genomic analyses and niche differentiation under actual environmental conditions can be uncoupled. Furthermore, investigations of the genomic and physiological properties of species across a range of ecological factor gradients are essential for understanding the adaptations between species in various environmental systems. Hence, further investigations and ecological experiments, such as *in situ* or microcosm studies with different environmental factor treatments, are necessary to understand the ecological adaptations of bacterial clusters.

## Conclusion

This study indicates that strain JHP9 represents a novel species of the genus *Brachybacterium* based on 16S rRNA analysis, genome-based phylogenetic analyses, and similarity matrix data using the ANI, AAI, and *is*DDH values. Physiological studies uncovered the diverse growth conditions and carbohydrate utilization of the novel strain, along with its chemotaxonomic phenotypes. The phylogenetic relationships among members of the genus *Brachybacterium* were determined to characterize the diversity of this genus. The versatile metabolic potential of the strain JHP9 to utilize various carbohydrates and lactic acid production enables its propagation and potential utilization in the food industry. In addition, this is the first study to report the high oxygen and glucose affinity of strain JHP9, its resistance to quinolone and phenicol antibiotics, and the prevalence of CRISPR systems in its genome, suggesting its versatility, adaptability, and opportunism. Taken together, future in-depth analyses of *Brachybacterium* spp. to uncover their interactions in various environments, genome plasticity, and impact on neighboring cells are required to completely understand their beneficial or harmful dynamics in the natural environment. Further metagenomics-based studies should be conducted to estimate their abundance across various ecosystems to elaborate on their adaptation versatility.

### Description of B. equifaecis JHP9 sp. nov. (L.n. Equus, horse; L. n. faex, -cis, yeast, feces; N.L. gen. n. equifaecis, from horse feces)

Cells of strain JHP9 are gram positive, non-motile, non-flagellate, and rod shaped, with a diameter of $0.8 \times 1.8$ µm. Colonies are cream colored, convex, smooth, glistening, and 0.5–1 mm in diameter on TSA incubated at 30°C for 7 d. It grows optimally at 30°C (growth range, 18–37°C), pH 7.0 (growth range, 6.0–9.0), and at a 0.5–1% (growth range 0–10%, wt/vol) salt concentration. Cell wall peptidoglycan of strain JHP9 contains amino acids, including *meso*-DAP, alanine, and glutamic acid. The major isoprenoid quinones and fatty acids were MK-7 and $C_{15:0}$ anteiso and $C_{19:0}$ cyclo $\omega 8c$, respectively. The main polar lipid profile comprised DPG, PG, and GL. The JHP9 strain is resistant to chloramphenicol and kanamycin. It hydrolyzes starch and DNA but not casein, cellulose, Tween 20, and Tween 80. The hydrolysis activity of $\beta$-glucosidase was positive but that of protease was negative. The strain is positive for catalase and negative for oxidase activity, indole formation, dissimilatory nitrate reduction, and urease activity. Fermentation activity was

positive for D-ribose, D-galactose, D-glucose, D-mannose, methyl α-D-glucopyranoside, esculin, D-cellobiose, D-lactose, D-trehalose, D-raffinose, and glycogen. The genome of strain JHP9 was 3.08 Mbp in size, with a 71.1% G + C content.

Based on phenotypic, genotypic, and phylogenetic analyses, it was proposed that strain JHP9 represents a novel species with the name *B. equifaecis* sp. nov., with the type strain being JHP9 (= KCTC 49746[T] = JCM 35094[T]).

## MATERIALS AND METHODS

### Isolation and morphological characterization

The *B. equifaecis* JHP9 strain was obtained from the fecal samples of pasture breeding horses in Jeju Island, Republic of Korea (33°26'46.9"N, 126°33'50.7"E). In a 50-mL plastic tube, 10-g fecal samples were collected from the center of a horse fecal ball and transferred to the laboratory on ice. Nine samples (0.1 g each) were inoculated into 1-mL sterilized phosphate-buffered saline and diluted 10 times to a $10^{-5}$ dilution. Subcultures of $10^{-3}$, $10^{-4}$, and $10^{-5}$ dilutions were conducted on Lactobacilli MRS agar (BD Difco, Franklin Lakes, NJ, USA) by spreading 500 µL of each dilution and incubating them at 30°C for 72 h under ambient air conditions. After several subcultures, single colonies based on colony morphology and color were selected to inoculate onto fresh MRS agar at 30°C. Strain JHP9, isolated from the $10^{-4}$-diluted fecal sample, was inoculated onto MRS agar plates and incubated at 30°C for 48 h. The isolated strain was regularly cultured on TSA (Oxoid, Hampshire, UK) plates. Gram staining was performed using a Gram staining kit (BD Difco) according to the manufacturer's instructions, and colony morphology was examined microscopically. The shape and size characteristics of the cells were confirmed using TEM (Tecnai G2 Spirit Twin, FEI; as installed at the Korea Basic Science Institute) after using 1% phosphotungstic acid for negative staining. Stock cultures were stored in 20% glycerol at −80°C until characterization.

### Phylogenetic analysis for strain identification

The isolated strain was identified using molecular phylogenetic analysis based on approximately 1,453 bp of the 16S rRNA gene sequences. For this purpose, the genomic DNA of *B. equifaecis* JHP9 was extracted using a genomic DNA extraction kit (Bioneer, Korea). The 16S rRNA gene was PCR amplified using universal primers 27F (5′-AGA-GTTTGATCMTGGCTCAG-3′; *E. coli* position 8–27) and 1492R (5′-TACGGYTACCTTGTTAC-GACTT-3′; *E. coli* position 1492–1510) (59) and the purified PCR product sequenced by Macrogen Co. Ltd. (Republic of Korea) using an ABI 3730xl DNA Analyzer (Thermo Fisher Scientific, Waltham, MA, USA) with the BigDye Terminator v3.1 Cycle Sequencing Kit (Thermo Fisher Scientific) following the manufacturer's protocols. Sequencing was conducted on each template using 518F (5′-CCAGCAGCCGCGGTAATACG-3′), 785F (5′-GGATTAGATACCCTGGTA-3′), 800R (5′-TACCAGGGTATCTAATCC-3′), and 907R (5′-CCGTCAATTCMTTTRAGTTT-3′) primers to cover the entire region of the 16S rRNA gene.

An almost-complete 16S rRNA gene sequence (approximately 1.5 kbp) was obtained by assembling the sequences using the BioEdit v.7.2.6 software (60) with the CAP contig assembly program and comparing it with the 16S rRNA gene sequence extracted from the whole genome sequence of strain JHP9 (see below). Then, the sequence was compared with the 16S rRNA gene sequences of related taxa obtained from the GenBank database and the EzBioCloud server (https://www.ezbiocloud.net). The 16S rRNA gene sequences of representative members of *Brachybacterium* and an outgroup bacterium, *Sanguibacter suarezii* NBRC16159[T], which has <90% 16R rRNA gene similarity with strain JHP9, were aligned with that of the JHP9 strain using SILVA (http://www.arb-silva.de/aligner), where the secondary structure of the rRNA gene was considered (61). Phylogenetic tree construction was accomplished using maximum likelihood, neighbor-joining, and maximum parsimony methods implemented in the MEGA11 program (62).

Kimura's two-parameter model (63) was employed to calculate phylogenetic distances, and bootstrap analysis was conducted based on 1,000 resampled data sets.

Based on the 16S rRNA gene sequences, the strain shares a high sequence similarity (>97%) with *B. nesterenkovii* JCM11648[T], *B. huguangmaarense* JCM30544[T], *B. horti* KCTC39563[T], and *B. rhamnosum* KCTC9917[T]. Therefore, these type strains were obtained from the Japanese Collection of Microbes (JCM) (JCM11648[T] and JCM30544[T]) and the Korean Collection for Type Cultures (KCTC) (KCTC39563[T] and KCTC9917[T]) to compare their phenotypic characteristics with those of the isolated strain.

## Physiological and biochemical characterization

The temperature, pH, and NaCl ranges for growth were determined in triplicate in tryptone soya broth (TSB; Oxoid). Growth temperature, NaCl concentration, and pH ranges were tested over the ranges of 4–45℃ (4, 10, 15, 18, 25, 30, 37, 42, and 45℃) at pH 7 for temperature determination, 0–25% (wt/vol) NaCl (0, 0.5, 1, 2, 3, 6, 7, 8, 9, and 10%) at 30℃ and pH 7 for NaCl concentration analysis, and pH 4–10 (at intervals of 0.5 pH units) for adjusted final pH using NaOH (1N) and HCl (1N) at 30℃. Four different buffers were used in the pH response analysis (final concentration, 10 mM): homopiperazine-1,4-bis (2-ethanesulfonic acid) (pH 4.0–5.0), 2-(N-morpholino)ethanesulfonic acid (pH 5.0–6.5), 1,3-bis[tris(hydroxymethyl)methylamino]propane (bispropane, pH 7.0–8.5), and 3-(cyclohexylamino)-1-propanesulfonic acid (pH 9.0–10.0). Bacterial growth was evaluated by measuring culture absorbance at 600 nm using a DS-11 spectrophotometer (DeNovix, Wilmington, DE, USA) at the Bio-Health Materials Core-Facility (Jeju National University, Republic of Korea) after incubation for 24 h, 48 h, and a week at different temperatures, salinities, and pH conditions.

Catalase activity was determined by observing bubble production in 3% (wt/vol) hydrogen peroxide solution (64), and oxidase activity was confirmed by observing color transitions to a deep purple or blue after adding drops of 1% (wt/vol) tetramethyl-*p*-phenylenediamine solution (bioMérieux, France) (65). Bacterial enzyme activities were examined using API ZYM and API 20 NE test strips (bioMérieux). API 20 NE test strips were also used to assess the ability of bacteria to assimilate carbohydrates. The bacterial ability to ferment and produce acid from various carbohydrates was tested with API 50 CHL test strips (bioMérieux) using API 50 CHB/E media with mineral oil under anaerobic conditions in an anaerobic chamber (Coy Laboratory Products, Grass Lake, MI, USA), according to a previously described method (66). Starch degradation and Tween 20 and Tween 80 hydrolysis were tested as previously described (67, 68). DNA hydrolysis was determined by culturing the cells on DNase Test Agar containing methyl green (BD Difco). A zone of clearing around the bacterial colonies indicated positive activity.

Additionally, six sugar sources (glucose, sucrose, galactose, arabinose, lactose, and mannose) were used at a ratio of 1% to determine fermentation activity in the strains JHP9, *B. nesterenkovii* JCM11648[T], and *B. huguangmaarense* JCM30544[T], along with two control LAB strains of *L. delbrueckii* subsp. *bulgaricus* KCTC3769[T], under anaerobic conditions (see above). Briefly, 0.001% TSB media were prepared for *Brachybacterium* and LAB strains using phenol red and minimal media, called artificial freshwater medium (AFM) (15). Subsequently, bacteria were inoculated into each medium separately containing the six sugars. Incubation was performed at 30℃ for 2 wk anaerobically, and after incubation, lactic acid production was observed by detecting a color change in the media. Alterations in pH and lactic acid production tended to cause color changes to the phenol red. Detailed methodology is provided in the Supplementary Information. In addition, lactic acid production was observed when 5-mL growth culture of strains JHP9, JCM 11648[T], JCM 30544[T], and KCTC 3769[T] was supplemented with sugars under anaerobic conditions. Finally, the lactic acid concentration was measured with an EZ-Lactate Assay kit (DoGenBio, Korea) and a SpectraMax iD5 microplate reader (Molecular Devices, San Jose, CA, USA) at the Bio-Health Materials Core-Facility (Jeju National University) according to the manufacturer's instructions.

The disk diffusion method was used to determine antibiotic susceptibility patterns of the bacteria (69). The following antibiotics were tested in the present study: ampicillin (10 µg per disk), streptomycin (10 µg per disk), chloramphenicol (30 µg per disk), ciprofloxacin (5 µg per disk), gentamicin (10 µg per disk), kanamycin (30 µg per disk), and tetracycline (30 µg per disk). These antibiotics were selected on the basis of their common use for therapeutic and prophylactic purposes in horse (https://www.woah.org/app/uploads/2021/03/oie-list-antimicrobials.pdf). Antibiotic susceptibility was defined based on breakpoints provided by the Clinical and Laboratory Standards Institute guidelines.

## Chemotaxonomic analyses

Chemotaxonomic analyses were performed for peptidoglycan, fatty acids, polar lipids, and menaquinone after culturing cells for 48 h on TSA (1% [wt/vol] NaCl, pH 7) at 30°C. Briefly, cell wall peptidoglycan of strain JHP9 was hydrolyzed and extracted using 6 N HCl at 121°C for 15 min and analyzed on a cellulose TLC plate according to a previously described method (70). Cellular fatty acids were extracted from strains JHP9, JCM11648[T], and JCM30544[T] via gas chromatography (HP7890 GC-FID; Agilent Technologies, Santa Clara, CA, USA) according to protocols of the Sherlock Microbial Identification System (MIDI; version 6.1) and the TSBA6 database (71). Identification and quantification of the extracted fatty acids were performed using the Sherlock MIDI system and standard MIS library generation software (version 6.3; Microbial ID Inc., Newark, DE, USA). Polar lipids were extracted using a chloroform-methanol solution (2:1, vol/vol) according to a previously described method (72) and analyzed using two-dimensional TLC. Furthermore, aminolipids, phospholipids, and glycolipids were identified using ninhydrin, molybdenum blue spray reagent, and α-naphthol-sulfuric acid, respectively, by heating at 110°C for 15 min. The isoprenoid quinones of strain JHP9 were identified with chloroform: methanol (2:1, vol/vol) using high-performance liquid chromatography (HPLC 2487; Waters Corporation, Milford, MA, USA) with a reversed-phase column, as described previously (73, 74). Detailed chemotaxonomic analysis methodology is provided in the Supplementary Information.

## Genome sequencing and analyses

Genomic DNA was extracted from the isolated bacteria using a QIAamp DNA Mini Kit (Qiagen, Germany) according to the manufacturer's instructions. We previously described the draft genome sequence of the novel JHP9 genome (75). After quantification, the genomic DNA was sequenced at Macrogen, Inc. (Seoul, South Korea), using the Illumina HiSeqX platform. Raw reads were verified for quality using FastP v0.23.1 (76), followed by assembly using Spades v1.22 (77) with the default parameters. The assembled genome was validated using a self-mapping strategy and BUSCO analysis (78). BUSCO analysis is performed to evaluate the genome assembly based on evolutionarily informed expectations of gene content. The filtered reads were aligned against the assembled genome, and their insert sizes were estimated for validation.

Genome annotation was performed with Prokka v1.13 (79) using default parameters and the Pfam database (80) with a BLAST identity >80%. In addition, CAZymes involved in the synthesis, metabolism, and recognition of complex carbohydrates were predicted using the CAZymes Analysis Toolkit (81) and dbCAN HMM version 4.0 (82), using the CAZy database. Secondary metabolite biosynthetic gene clusters were identified using antiSMASH 6.0 software (83) with strict detection. Genome sequences were further screened for ARGs using AMRFinderPlus (84) with default parameters. Potential virulence genes were identified using a virulence factor database (VFDB) with the VFDB core data set of proteins associated with experimentally verified virulence factors (85). MGEs in the genomes, including insertion sequence elements and transposons, were detected by using IseScan (86) and TEfinder (87), respectively. CRISPR systems in *Brachybacterium* genomes were identified using the CRISPR comparison toolkit (88), and prophages were identified with Phispy v4.2.19 (89) without HMM searches. Additionally, functional

assignment through homology searching against the specialized Kyoto Encyclopedia of Genes and Genomes database (90) was performed to determine the metabolic potential of the *Brachybacterium* genomes. All available, nearly-complete genomes of the *Brachybacterium* genus were obtained from the NCBI genome database (http://www.ncbi.nlm.nih.gov/genome/) and used for comparative analyses.

Genome distinctiveness was evaluated using genome-relatedness parameters, including ANI, AAI, and *is*DDH. The ANI and AAI calculations were performed using FastANI (91) and CompareM (https://github.com/dparks1134/CompareM), respectively. The *is*DDH values were calculated with the Genome-to-Genome Distance Calculator version 2.1 (http://ggdc.dsmz.de/distcalc2.php), using the recommended BLAST+ alignment (17).

Six *Brachybacterium* genomes were obtained from the NCBI genome database for comparative analysis with *B. equifaecis* JHP9. These genomes were chosen based on their similarity to *B. equifaecis* JHP9, as determined through phylogenetic analysis using 16S rRNA gene sequences. Additionally, we considered genome completeness of over 90%, which was assessed using BUSCO analysis (Table S1). Moreover, 163 complete representative genomes within the order *Micrococcales*, to which *Brachybacterium* belongs, were downloaded from the NCBI genome database for the phylogenetic clustering of *B. equifaecis* JHP9 within this order (Supplementary File 1).

Pangenome analysis of the available *Brachybacterium* and JHP9 genomes was performed using the Bacterial Pan Genome Analysis tool (BPGA) version 1.0 (92) with sequence identity ≥50% and an E value ≤$1.0 \times 10^{-5}$. Core, accessory, and unique genes were classified into orthologous groups using the USEARCH clustering algorithm (93). The *Brachybacterium* core genome sequences obtained from BPGA analysis were aligned against the NCBI RefSeq database (94) using BLAST with an identity cutoff of 50% to determine the closely associated genomes within the *Actinomycetota* phylum. Then, the closely associated 163 complete genomes (Supplementary File 1) within the order *Micrococcales* of the phylum *Actinomycetota* were downloaded from NCBI genome downloading scripts (available at https://github.com/kblin/ncbi-genome-download). These and the JHP9 genomes were subjected to alignment-free composition vector-based phylogenetic analyses using the CVTree4 program (95) with K = 6, and the resulting phylogenetic tree visualized and edited using iTOL v.6.5.2 (96).

## Determining kinetic properties

Kinetic properties were quantified by measuring the glucose- and oxygen-dependent oxygen uptake in an MR system (Unisense, Denmark) equipped with a PA 2000 picoammeter and an OX-MR oxygen microsensor (500-μm-diameter tip; Unisense, Denmark), polarized continuously for at least 24 h before use (15). MR experiments were conducted with each *Brachybacterium* strain under the respective cultivation conditions. To concentrate the biomass for MR, actively growing cells were collected from 5-mL culture by centrifugation ($8{,}000 \times g$, 10 min, 28°C). The concentrated biomass was washed twice with fresh AFM and resuspended in AFM for the MR experiments. The culture biomass was incubated in a water bath set to the experimental temperature prior to being transferred to a 2-mL glass MR chamber with a stir bar. All MR experiments were performed with 300 rpm stirring at 28°C. Small culture volumes were frequently used for glucose measurements during the experiment to confirm the stoichiometric conversion of oxygen in the glucose-dependent oxygen uptake experiments. The glucose concentration was measured using a Glucose Colorimetric Detection Kit (Thermo Fisher Scientific) according to the manufacturer's instructions. For oxygen-dependent oxygen uptake, 2 mM glucose was added to the bacterial cell-suspended chamber. Cell abundance was determined via qPCR using the bacterial primers, 518F-786R, as described by Jung et al. (97). The cell numbers used for the MR experiments were ~$1.7 \times 10^8$ for strain JHP9, ~$9.1 \times 10^8$ for *B. nesterenkovii*, ~$6.8 \times 10^7$ for *B. huguangmaarense*, ~$4.9 \times 10^8$ for *B. horti*, and ~$6.2 \times 10^8$.

## Data and statistical analysis

Resulting data from the physiological and chemotaxonomic analyses were analyzed descriptively to compare the characteristics of strain JHP9 with those of the reference strains. Statistical analyses were performed using the R computing environment (http://www.R-project.org/) and SigmaPlot 11.0 (Systat Software Inc., San Jose, CA, USA). Differences between lactic acid production values were analyzed using one-way analysis of variance (ANOVA). The significance level (α) for the ANOVA was 0.05. Tukey's comparison of means was used to identify the significant condition effects. The heatmaply package in R (https://cran.r-project.org/web/packages/heatmaply/index.html) was used to visualize genome similarity matrices.

## ACKNOWLEDGMENTS

This research was supported by the Research Institute for Basic Sciences of Jeju National University (2019R1A6A1A10072987), funded by the Ministry of Education, and by the National Research Foundation of Korea (NRF-2021R1C1C1008303 and NRF-2022R1A4A503144711), funded by the MSIT.

## AUTHOR AFFILIATIONS

[1]Research Institute for Basic Sciences (RIBS), Jeju National University, Jeju, South Korea
[2]Department of Marine Life Science, Jeju National University, Jeju, South Korea
[3]Interdisciplinary Graduate Programme in Advance Convergence Technology and Science, Jeju National University, Jeju, South Korea
[4]Mineral Resources Research Division, Korea Institute of Geoscience and Mineral Resources, Daejeon, South Korea
[5]Department of Biological Sciences and Biotechnology, Chungbuk National University, Cheongju, South Korea
[6]Department of Science Education, Jeju National University, Jeju, South Korea
[7]Jeju Microbiome Center, Jeju National University, Jeju, South Korea

## AUTHOR ORCIDs

Adeel Farooq http://orcid.org/0000-0002-0956-8773
Man-Young Jung http://orcid.org/0000-0002-5244-5197

## FUNDING

| Funder | Grant(s) | Author(s) |
|---|---|---|
| Jeju National University (JNU) | 2019R1A6A1A10072987 | Man-Young Jung |
| National Research Foundation of Korea (NRF) | NRF-2021R1C1C1008303 | Man-Young Jung |
| Ministry of Science and ICT, South Korea (MSIT) | NRF-2022R1A4A503144711 | Man-Young Jung |

## AUTHOR CONTRIBUTIONS

Adeel Farooq, Conceptualization, Formal analysis, Investigation, Methodology, Software, Visualization, Writing – original draft, Writing – review and editing | Myunglip Lee, Conceptualization, Data curation, Investigation, Methodology, Visualization, Writing – original draft, Writing – review and editing | Saem Han, Investigation, Methodology, Resources | Gi-Yong Jung, Methodology, Writing – review and editing | So-Jeong Kim, Data curation, Methodology, Validation, Writing – review and editing | Man-Young Jung, Conceptualization, Funding acquisition, Project administration, Resources, Supervision, Writing – review and editing

## DATA AVAILABILITY

The GenBank/EMBL/DDBJ accession number for the 16S rRNA gene is OL468816, whereas the NCBI GenBank accession number for the whole-genome sequence of JHP9 is JAKNCJ010000000. Strain JHP9 has been deposited in the KCTC (KCTC 49746) and the JCM (JCM 35094).

## ADDITIONAL FILES

The following material is available online.

### Supplemental Material

**Data Set S1 (Spectrum05048-22-s0001.xlsx).** Genome information for phylogenetic analysis in Fig. 1.
**Data Set S2 (Spectrum05048-22-s0002.xlsx).** CRISPR-cas types, terminal oxidases, MGEs, CAZYmes, and stress and antibiotic resistance genes.
**Text S1 (Spectrum05048-22-s0003.pdf).** Supplemental methods, tables, and figures.

### Open Peer Review

**PEER REVIEW HISTORY (review-history.pdf).** An accounting of the reviewer comments and feedback.

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
