## [Reviewer comments · Microbiology Spectrum]

Microbiology Spectrum

Kinetic, Genomic, and Physiological Analysis Reveals Diversity in the Ecological Adaptation and Metabolic Potential of *Brachybacterium equifaecis* sp. nov. Isolated from Horse Feces

ADEEL FAROOQ, Myunglip Lee, Saem Han, Gi-Yong Jung, So-Jeong Kim, and Man-Young Jung

Corresponding Author(s): Man-Young Jung, Jeju National University

Review Timeline:

Submission Date:	December 13, 2022
Editorial Decision:	May 18, 2023
Revision Received:	July 18, 2023
Accepted:	July 31, 2023

Editor: Jeffrey Gralnick

Reviewer(s): Disclosure of reviewer identity is with reference to reviewer comments included in decision letter(s). The following individuals involved in review of your submission have agreed to reveal their identity: Nagaraju Indugu (Reviewer #1)

Transaction Report:

DOI: <https://doi.org/10.1128/spectrum.05048-22>

May 18, 2023

Prof. Man-Young Jung
Jeju National University
Department of Science Education
Jeju-si, Jeju-do
Korea (South), Republic of

Re: Spectrum05048-22 (**Kinetic, Genomic, and Physiological Analysis Reveals Diversity in the Ecological Adaptation, and Metabolic Potential of a Novel Species, *Brachybacterium equifaecis***)

Dear Prof. Man-Young Jung:

Apologies for the extended delay and thank you for submitting your manuscript to Microbiology Spectrum. As you will see both reviewers have provide several points for your consideration on a variety of issues ranging from statistics to organization of the manuscript. Reviewer 1 suggests a revision to the title, which I will leave up to your discretion (I think your title is sufficiently descriptive of the study as is).

Link Not Available

Sincerely,

Jeffrey Gralnick

Senior Editor, Microbiology Spectrum

Journals Department
Reviewer comments:

Reviewer #1 (Comments for the Author):

I have given specific comments and suggestions in my report

Reviewer #2 (Comments for the Author):

The authors 1) presented a discovery of a new *Brachy bacterium* species and its physiological and phylogenetic characterization in comparison with close relatives, 2) conducted comparative genomic analysis among *Brachy bacterium* species to probe their functions in terms of niche adaptation and differentiation, and 3) performed kinetic analysis (glucose and oxygen uptake) to test some of their genomic findings.

The description of the new species is well done, and this reviewer also appreciates the inclusion of multiple *Brachy bacterium* species for comparative analysis. This provides rich data and a solid foundation for further studies of the ecophysiology of this genus.

Major comments:

The manuscript can benefit from restructuring its presentation (results & discussions) to make it flow better: 1) sequencing shows JHP9 is a *Brachy bacterium* species -> 2) characterize its basic physiology in comparison to its relatives -> sequence its genome and comparative genomics -> experimental verification of some of the key genomic features relevant to niche adaptation.

The isolate came from horse feces, but that niche is completely ignored - contrary to the authors' goal in studying niche differentiation.

The authors appear undecided on what their priorities are - they did *in silico* and experimental comparisons for multiple species, but their discussions revolved more or less around *B. equifaecis*. Focusing on the latter is completely fine, but the writing needs more restructuring - center primarily on the describing *B. equifaecis*, and then show examples of niche adaptation (not differentiation) as a secondary aim. Alternatively, it's great (and stronger) if the authors choose to continue to pursue niche differentiation, because they already have the comparative data. Here, the authors should view and discuss their results in the context of diverse niches.

Other comments:

Title: why not end the title with *Brachy bacterium equifaecis* sp. nov.?

63, rephrase - They are rarely isolated from humans, but a recent case report documented...

66, rephrase - Isolation of *Brachy bacterium* strains...

69, delete 'using...genera.'

69-71, rephrase - it reads as if the authors are talking about *Brachy bacterium*, but the citations are about other bacteria. If *Brachy bacterium* is heterotrophic, state that first before using other heterotrophic bacteria as examples.

73-74, delete 'however' and make the sentence concise. Try - features contributing to niche differentiation remain to be identified.

86, Rephrase - characterizing one strain does not build a strong case for niche differentiation of a genus. However, the authors did experiments on more than one strain - take credit for that.

89, state the N - how many strains are used here

90-96, this is not introduction and belongs to results and conclusions instead.

104, what's the gas phase for the cultivation?

111, microscopically

121 and 124, spell out the primer sequences and cite the relevant papers.

134, has the authors' strain been deposited into culture collection centers? If so, please provide proof.

166, why were the API 50 tests not carried out anaerobically?

285-310, this section is a good description of JHP9, but not obvious what sets it apart from other strains.

293, hydrolyzed DNA? A DNase test is not mentioned anywhere in the text.

335, the manuscript will flow better if this section goes first above all other results.

365-368, physiology data should be taken into account as well.

373-376, what about the other *Brachy bacterium* genomes? Does this support or refute your and others' experimental results?

388-391, Does this support or refute your and others' experimental results?

410, delete extra space before '.'

409-421, this paragraph should go first for this section. Then, go into the other paragraphs to present the examples of 1) shared core functions - would these support a heterotrophic and versatile lifestyle? 2) accessory functions - would these support specific niche differentiation? 3) unique functions - would these support niche adaptation?

423, how does this relate to the authors' intro about K vs S strategies?

424-437, citations when referring to others' work.

439, this section should discuss first the discovery and characterization of a novel species. It should also be rewritten depending on the author's priorities - niche adaptation or niche differentiation.

442-444, it's not obvious from the results what set JHP9 apart from other strains, and what is considered 'broad range'.

446, has it been detected in other environments beside horse feces? An alternative explanation is these features (particularly when not shared with other *Brachy bacterium*) are there for niche adaptation in the horse intestine.

447, delete 'technically'

451, physiological

470, what are the inhabiting environments?

495, conversely

496, The genes may well be expressed but perhaps, there is a dose effect undetected by the authors' assays. Clinical breakpoints are species-specific, the MIC for the same antibiotic can vary among pathogens. It's also not black and white - between resistant and susceptible, there is also intermediate. Were the horses being treated regularly with antibiotics and what kinds?

582, potential utilization... unless it's already been used in food industry.

Table 1 - any comments on the high GC content of Brachybacterium?

Figure 1A - highlight the authors' strain in bold, and underline the species selected for comparative analysis. Figure legends have A and B mixed up.

Figure 2A - color keys?

Figure 2B - unit for Y axis?

Figure 3B - why the big variation for E. coli?

Figure 4 - nice figure but a summary of the main takeaways should be included in the figure legends.

Suppl. Fig. S5 - the oxygen data is excellent, but the glucose data is patchy. In particular, this reviewer does not feel confident about the glucose data for B. nesterenkovi and B. huguangmaarensis.

Staff Comments:

Preparing Revision Guidelines

Please return the manuscript within 60 days; if you cannot complete the modification within this time period, please contact me. If you do not wish to modify the manuscript and prefer to submit it to another journal, please notify me of your decision immediately so that the manuscript may be formally withdrawn from consideration by Microbiology Spectrum.

Adeel Farooq *et al* characterized the *Brachybacterium equifaecis* species isolated from horse feces. The authors had already published the genomic data of this study as a draft genome article. In this study the authors reported additional data with some additional analysis including physiological, Kinetic and genomic (16s rRNA & whole genome shotgun sequencing) analysis and compared with similar *Brachybacterium* species. Overall, the findings of this study looks curious. However, some specific areas need to be addressed before publishing this manuscript in the mSpectrum journal. The manuscript needs to be organized for sequencing order as per the experiment conducted with separate subsection or paragraph for each analysis. An editorial correction for English and proof reading are required. In some places the results were not well written they just provided the tables or figures e.g the antibiotic sensitivity test and the results of KEGG analysis. The data should be presented in results section with an appropriate statistics or descriptive analysis, the authors introduced the abbreviation but not defined on its first appearance. A separate section is required for the selection criteria for reference genomes used for comparative and/or for pan-genome analysis. Statistical analysis or data analysis section is missing.

Title:

Please include “*Brachybacterium equifaecis* is isolated from horse feces” in the title.

Abstract:

I see some discussion points in the abstract. Authors should have highlighted the approach for comparison with other strains. Please organize the abstract with brief introduction, brief methodology and important results and main conclusions.

Introduction:

Although the introduction is written well the justification for the isolation of the *Brachybacterium equifaecis* strain from horse is missing. For example, the importance of this strain in the horses needs to be addressed. The hypothesis of the study is missing. Authors should have introduced why they have used comparative analysis or pan-genome analysis.

Ln 94: The sentence is in the future tense. Generally, the introduction ends with hypothesis and appropriate objectives. Please revise accordingly.

Materials and Methods:

Ln102: Please include additional information for the collection of samples e.g how were the sample collected is it collected from center or surface of the fecal ball or from rectum or feces collected from the floor?

Ln109: Please use any other word for “routinely”

Ln121: Please provide the nucleotide sequences of the primers used in this study.

Ln124-125: “using 518F, 785F, 800R, and 907R” This is not clear. Also please provide which sequencing platform used.

Ln125:131- More details are required.

Were you used all the sequences from the database or used only Brachybacterium sequences for sequences alignment. 2) What algorithm was used for pairwise alignment? 3) You have mentioned 3 different tree construction methods. Report the correct phylogeny construction method. 4) It is not clear what is outgroup organism.

Ln153: provide reference for catalase activity methodology.

Results:

Ln 308: More details are required in the results section of antibiotic sensitivity test. Please include appropriated resultant data in parenthesis.

Ln 323-Ln333- This looks like interpretation of the results can be moved into discussion section.

Ln340-Ln342- This is repetition at Ln 365-368, so these lines can be removed here.

Ln353: Is table S4 correctly cited? It should be Table S5.

Ln389-390- I believe the values in the parenthesis are number of CAZYmes detected across the 7 strains.

Discussion

Please include the any limitations of the current study.

Figure 4 appear in the discussion. The corresponding results not described in the introduction.

Ln496: Authors stated “genes were not expressed” while no expression analysis performed in this study.

Figures:

Figure 1

Switch the legends 1a) and 1b). Not matching with figure 1a and 1b. In Figure 1A, highlight the strain JHP9 with bold. What does the numbers (e.g 100/100/100) near the node indicates. Please describe in the figure legend.

Supp Figure S1

Represent the size of the black scale bars in Figure S1 legend. Also, describe the two panels.

Response to Reviewers' comments

We would like to express our gratitude to the reviewers for their diligent examination of the manuscript and for identifying crucial areas for improvement. Please find below our responses to the remarks of the Reviewers as inserts (in blue):

Comments from reviewer(s): Reviewer #1:

Adeel Farooq *et al* characterized the *Brachybacterium equifaecis* species isolated from horse feces. The authors had already published the genomic data of this study as a draft genome article. In this study the authors reported additional data with some additional analysis including physiological, Kinetic and genomic (16s rRNA & whole genome shotgun sequencing) analysis and compared with similar *Brachybacterium* species. Overall, the findings of this study look curious. However, some specific areas need to be addressed before publishing this manuscript in the mSpectrum journal. The manuscript needs to be organized for sequencing order as per the experiment conducted with separate subsection or paragraph for each analysis.

Re) We are thankful to the reviewer for recognizing the findings of our study. Furthermore, we concur with the reviewer's suggestion to enhance the organization of the manuscript by including a description of each analysis. Please check the revised manuscript, which has been restructured.

An editorial correction for English and proof reading are required.

Re) We thank the reviewer for this point. As evidence of our commitment to improving the quality of our manuscript, we have included an English correction and proofreading certificate at the end of this document.

In some places the results were not well written they just provided the tables or figures e.g the antibiotic sensitivity test and the results of KEGG analysis.

Re) Thank you for alerting us to this point. Below, we present the revised versions of the results for the KEGG analysis and susceptibility test:

" Functional assignment of the pangenome revealed variations in the distribution of genes within the different metabolic fractions (Figure 2C). The core genome exhibited a prominent presence in essential processes, such as amino acid metabolism, translation, nucleotide metabolism, replication and repair, energy metabolism, folding, sorting and degradation, cell growth and death, and transcription. In contrast, the accessory genome showed significant representation of functions related to carbohydrate metabolism, cellular community, terpenoid and polyketide metabolism, drug resistance, and xenobiotic biodegradation and metabolism. In contrast, the unique genome demonstrated substantial representation of functions associated with membrane

transport, xenobiotic biodegradation and metabolism, lipid metabolism, biosynthesis of secondary metabolites, glycan biosynthesis and metabolism, cell motility and transport, and catabolism (Figure 2C).” (Lines 192-202)

“Based on the susceptibility test results, all three Brachy bacterium species were susceptible to ampicillin, gentamicin, streptomycin, and tetracycline. However, these strains were resistant to kanamycin. Additionally, strain JHP9 displayed resistance to chloramphenicol, whereas the remaining strains demonstrated susceptibility to it (Table S4).” (Lines 182-185)

The data should be presented in the results section with an appropriate statistics or descriptive analysis, the authors introduced the abbreviation but not defined on its first appearance.

Re) We extend our gratitude to the reviewer for bringing this to our attention. In response, we have corrected the abbreviation. And we have included a comprehensive descriptive analysis of the results with an appropriate statistic in each result. We also added the data and statistical analysis part in the material and methods section.

A separate section is required for the selection criteria for reference genomes used for comparative and/or for pan-genome analysis.

Re) In response to the reviewer's comment, we appreciate the suggestion to include a separate section detailing the selection criteria for reference genomes used in the comparative and/or pan-genome analysis. We acknowledge the importance of providing transparency in our methodology and will incorporate a dedicated section that outlines the criteria employed for selecting reference genomes in our study. This additional information will enhance the clarity and comprehensiveness of our research. Thank you for this valuable suggestion. We have added a separate paragraph on the selection criteria of publicly available genomes as *“Six Brachy bacterium genomes were obtained from the NCBI genome database for comparative analysis with B. equifaecis JHP9. These genomes were chosen based on their similarity to B. equifaecis JHP9, as determined through phylogenetic analysis using 16S rRNA gene sequences. Additionally, we considered genome completeness of over 90%, which was assessed using BUSCO analysis (Table S1). Moreover, 163 complete representative genomes within the order Micrococcales, to which Brachy bacterium belongs, were downloaded from the NCBI genome database for the phylogenetic clustering of B. equifaecis JHP9 within this order (Supplementary File 2).” (Lines 655-662)*

The statistical analysis or data analysis section is missing.

Re) We appreciate the reviewer for highlighting the absence of a statistical analysis or data analysis section in our manuscript. As described above, we added the data and statistical analysis part in the material and methods section. Thanks again.

Title

Please include “*Brachy bacterium equifaecis* is isolated from horse feces” in the title.

Re) We thank the reviewer for the suggestion to include the isolation source in the manuscript title. Taking this suggestion into consideration, we have incorporated this important information in the title as “*Kinetic, Genomic, and Physiological Analysis Reveals Diversity in the Ecological Adaptation, and Metabolic Potential of Brachy bacterium equifaecis sp. nov. Isolated from Horse Feces.*”

Abstract

I see some discussion points in the abstract. Authors should have highlighted the approach for comparison with other strains. Please organize the abstract with brief introduction, brief methodology and important results and main conclusions.

Re) We appreciate the reviewer's feedback regarding the organization of the abstract. We acknowledge that there are discussion points included in the current abstract and understand the importance of clearly presenting the approach for comparison with other strains. We restructured the abstract to include a brief introduction, a concise methodology overview, key results, and main conclusions.

Introduction

Although the introduction is written well the justification for the isolation of the *Brachy bacterium equifaecis* strain from horse is missing. For example, the importance of this strain in the horses needs to be addressed.

Re) We greatly appreciate the reviewer's comment regarding the need for a more comprehensive justification for isolating the *Brachy bacterium equifaecis* strain from horses. We have updated the manuscript by adding the source information below:

“The gut microbiota plays a vital role in the health, metabolism, and overall well-being of the host. Horses belong to a family of herbivorous mammals that possess a certain hindgut (cecum and colon) microbiota, which provide a substantial proportion of energy for horses through fermentation. Furthermore, equine gut microbiota contributes to essential physiological processes such as digestion, nutrient absorption, and immune system development (2, 3). The equine gut microbiota plays a vital role; however, available data and studies on horse microorganisms are limited. Therefore, isolating and characterizing novel microorganisms from the horse gut offers an opportunity to enhance our understanding of the microbial diversity, unique characteristics, and functional capacities associated with equine gut microbiota. Genomic versatility and the consequent physiological properties of novel strains regarding their adaptation to various ecosystems, including the horse gut, are also required.” (Lines 66-77)

The hypothesis of the study is missing. Authors should have introduced why they have used comparative analysis or pan-genome analysis.

Re) Thank you for pointing out the missing hypothesis in our manuscript. We rephrased the last paragraph of the Introduction as follows:

“Consequently, this study aimed to determine the physiological, kinetic, phylogenetic, and genomic properties of Brachybacterium species, particularly emphasizing those of a novel B. equifaecis JHP9 strain isolated from horse feces. We investigated the potential adaptability of Brachybacterium species to natural habitats by employing comparative genomics of seven Brachybacterium genomes and conducting physiological analyses using reference genomes. We focused on elucidating their metabolic potential, particularly their use of various carbohydrates, adaptive strategies, and cellular kinetics. In this study, we sought to enhance our understanding of Brachybacterium species and their ecological significance. Specifically, by characterizing the features of strain JHP9, we aimed to provide valuable insights into its adaptative capabilities, potential applications, and the environmental role for its host, the horse.” (Lines 104-113)

Ln 94: The sentence is in the future tense. Generally, the introduction ends with hypothesis and appropriate objectives. Please revise accordingly.

Re) We acknowledge the reviewer's comment regarding the use of the future tense in the sentence. To adhere to the standard structure of an introduction, we rephrased the sentence accordingly.

Materials and Methods

Ln102: Please include additional information for the collection of samples e.g how were the sample collected is it collected from center or surface of the fecal ball or from rectum or feces collected from the floor?

Re) Thank you for your valuable suggestion regarding the collection of samples. In this study, the samples were collected using a standardized protocol. Fecal samples were obtained specifically from the center of the fecal ball. By focusing on the center of the fecal ball, we aimed to obtain a sample that maximized the representation of the internal microbial population while minimizing external influences. The collection procedure involved wearing disposable gloves, using a sterile collection tube, and ensuring minimal contact with the external environment to avoid contamination. The samples were immediately transferred to sterile containers and transported to the laboratory under appropriate storage conditions to preserve the microbial composition. We added this information in the manuscript as *“In a 50 mL plastic tube, 10 g fecal samples were collected from the center of a horse fecal ball and transferred to the laboratory on ice.” (Lines 504-506)*

Ln109: Please use any other word for “routinely”

Re) Thank you for your suggestion, and we have revised the sentence as *“The isolated strain was regularly cultured on TSA (Oxoid, Hampshire, UK) plates.” (Lines 512-513)*

Ln121: Please provide the nucleotide sequences of the primers used in this study.

Re) We thank the reviewer for bringing this to our attention, and we have incorporated the nucleotide sequences of the primers used in our study in the revised manuscript.

Ln124-125: “using 518F, 785F, 800R, and 907R” This is not clear. Also please provide which sequencing platform used.

Re) We appreciate the reviewer's comment. We have revised and included the required information in the revised manuscript.

Ln125:131- More details are required. Were you used all the sequences from the database or used only *Brachybacterium* sequences for sequences alignment. 2) What algorithm was used for pairwise alignment? 3) You have mentioned 3 different tree construction methods. Report the correct phylogeny construction method. 4) It is not clear what is outgroup organism.

Re) We appreciate the reviewer's attention to these details, and we have made the necessary revisions to address these concerns. We have reflected the above-mentioned description in the revised manuscript in the following manner: “*An almost complete 16S rRNA gene sequence (approximately 1.5 kbp) was obtained by assembling the sequences using the BioEdit v.7.2.6 software (61) with the CAP contig assembly program and comparing it with the 16S rRNA gene sequence extracted from the whole genome sequence of strain JHP9 (see below). Then, the sequence was compared with the 16S rRNA gene sequences of related taxa obtained from the GenBank database and the EzBioCloud server (<https://www.ezbiocloud.net>). The 16S rRNA gene sequences of representative members of Brachybacterium and an outgroup bacterium, Sanguibacter suarezii NBRC16159T, which has <90% 16R rRNA gene similarity with strain JHP9, were aligned with that of the JHP9 strain using SILVA (<http://www.arb-silva.de/aligner>), where the secondary structure of the rRNA gene was considered (62). Phylogenetic tree construction was accomplished using maximum likelihood, neighbor-joining, and maximum parsimony methods implemented in the MEGA11 program (63). Kimura's two-parameter model (64) was employed to calculate phylogenetic distances, and bootstrap analysis was conducted based on 1,000 resampled datasets.*” (Lines 533-545)

Ln153: Provide reference for catalase activity methodology.

Re) Thank you for the point. We have now included the appropriate reference in the revised manuscript.

Results

Ln 308: More details are required in the results section of antibiotic sensitivity test. Please include appropriated resultant data in parenthesis.

Re) Thank you for the valuable comment regarding the antibiotic sensitivity test. We have already addressed this concern, and please see the response above. Many thanks again.

Ln 323-Ln333- This looks like interpretation of the results can be moved into discussion section.

Re) Thank you for the valuable feedback on the interpretation of our results. We concur that certain interpretations in the results section would be more appropriately presented in the

discussion section. We have modified and updated the lactic acid production results, so please check the updated result and discussion regarding the results. (Lines 258-270)

Ln340-Ln342- This is repetition at Ln 365-368, so these lines can be removed here.

Re) We appreciate the reviewer's observation regarding the repetition. We have removed the repetition in the MS.

Ln353: Is table S4 correctly cited? It should be Table S5.

Re) Thank you for the points. It has been corrected.

Ln389-390- I believe the values in the parenthesis are number of CAZYmes detected across the 7 strains.

Re) They indeed represent the number of CAZymes detected across the seven strains analyzed in our study. In the revised statement, we incorporated the count of *Brachybacterium* genomes (n=7) in which CAZymes were identified. Thanks.

Discussion

Please include the any limitations of the current study.

Re) We thank the reviewer for this point. Considering this, we have revised the manuscript to include a paragraph addressing the limitations of our study as follows:

“In this study, we comparatively characterized various Brachybacterium species, with a particular focus on B. equifaecis JHP9, to assess their ecological adaptation and metabolic potential. However, the physiological properties deduced from in vitro and genomic analyses and niche differentiation under actual environmental conditions can be uncoupled. Furthermore, investigations of the genomic and physiological properties of species across a range of ecological factor gradients are essential for understanding the adaptations between species in various environmental systems. Hence, further investigations and ecological experiments, such as in situ or microcosm studies with different environmental factor treatments, are necessary to understand the ecological adaptations of bacterial clusters.” (Lines 454-462)

Figure 4 appear in the discussion. The corresponding results not described in the introduction.

Re) Thank you for bringing this to our attention. We have incorporated these corresponding results in discussion section: *“Thus, the metabolic potential of the nove JHP9 strain was further explored to uncover the complete or nearly complete metabolic pathways encoded by its genome.. In-depth analysis revealed that its ABC transportation system consisted of various components, including oligosaccharides (9 ea), phosphates (4 ea), minerals (2 ea), metallic cations (2 ea), and monosaccharides (1 ea) (Figure 3). Furthermore, metabolic pathways for carbohydrates, such as maltose, cellobiose, starch, galactose, D-mannitol, trehalose, and mannose, were identified within the genome. Additionally, it exhibited the presence of the pentose phosphate pathway, terpenoid backbone biosynthesis, pyruvate metabolism, amino acid metabolism, GL*

metabolism, nucleotide sugar biosynthesis, and the metabolism of cofactors and vitamins. Genomic analysis of this bacterial strain revealed the presence of pathways related to oxidative phosphorylation, demonstrating its capacity for efficient energy production through aerobic respiration (Figure 3). Notably, similar metabolic potential patterns were identified in other Brachy bacterium genomes, except for variations in the presence of ABC transporters (Figure S6).” (Lines 204-216)

Ln496: Authors stated “genes were not expressed” while no expression analysis was performed in this study.

Re) We appreciate the reviewer's comment regarding the statement we made about the expression of antibiotic resistance genes. To clarify this point and elaborate the lack of phenotypic resistance we have revised the manuscript to explicitly state that “*Conversely, the JHP9 genome carried beta-lactam-resistance genes despite exhibiting susceptibility to the corresponding antibiotic when tested phenotypically. The lack of phenotypic resistance associated with the beta-lactam-resistance gene, blaIII, may be attributed to several factors. These factors include mutations or genetic variations within the gene or its regulatory elements, absence of specific inducers or repressors, and genetic regulatory mechanisms (38-40).*” **(Lines 365-370)**

Figures

Figure 1: Switch the legends 1a) and 1b). Not matching with figure 1a and 1b. In Figure 1A, highlight the strain JHP9 with bold. What does the numbers (e.g 100/100/100) near the node indicates. Please describe in the figure legend.

Re) Thank you for bringing these issues to our attention. In the revised manuscript, we made corrections to the figure legends and included information regarding the numbers adjacent to the branch nodes.

Supp Figure S1: Represent the size of the black scale bars in Figure S1 legend. Also, describe the two panels.

Re) Thank for the comment. We have added the size of the scale bar in Figure S1.

Comments from reviewer(s): Reviewer #2:

The authors 1) presented a discovery of a new *Brachybacterium* species and its physiological and phylogenetic characterization in comparison with close relatives, 2) conducted comparative genomic analysis among *Brachybacterium* species to probe their functions in terms of niche adaptation and differentiation, and 3) performed kinetic analysis (glucose and oxygen uptake) to test some of their genomic findings.

The description of the new species is well done, and this reviewer also appreciates the inclusion of multiple *Brachybacterium* species for comparative analysis. This provides rich data and a solid foundation for further studies of the ecophysiology of this genus.

Re) Thank you for your positive feedback and appreciation of our work. We are glad that you found the description of the new species and the inclusion of multiple *Brachybacterium* species for comparative analysis to be well done. By including multiple *Brachybacterium* species in our study, we aimed to provide a comprehensive and comparative analysis of their characteristics, which can serve as a solid foundation for further investigations into the ecophysiology of this genus. Once again, we appreciate your thoughtful review and supportive comments.

Major comments:

The manuscript can benefit from restructuring its presentation (results & discussions) to make it flow better: 1) sequencing shows JHP9 is a *Brachybacterium* species -> 2) characterize its basic physiology in comparison to its relatives -> sequence its genome and comparative genomics -> experimental verification of some of the key genomic features relevant to niche adaptation.

Re) We appreciate your suggestion to restructure the presentation of the manuscript to improve the flow. Based on your recommendations, we restructured the manuscript. Thanks!

The isolate came from horse feces, but that niche is completely ignored - contrary to the authors' goal in studying niche differentiation.

Re) We acknowledge the reviewer's comment regarding the niche from which the isolate was obtained. We have revised our discussion to include a brief description of the horse feces niche and its potential relevance to our findings.

The authors appear undecided on what their priorities are - they did *in silico* and experimental comparisons for multiple species, but their discussions revolved more or less around *B. equifaecis*. Focusing on the latter is completely fine, but the writing needs more restructuring - center primarily on the describing *B. equifaecis*, and then show examples of niche adaptation (not differentiation) as a secondary aim. Alternatively, it's great (and stronger) if the authors choose to continue to pursue niche differentiation, because they already have the comparative data. Here, the authors should view and discuss their results in the context of diverse niches.

Re) We value the reviewer's insightful comment regarding the need for improved organization and clarity in our manuscript. We agree that focusing primarily on *B. equifaecis* will create a

more cohesive and coherent narrative for our study. In the revised manuscript, we have restructured our writing to highlight the specific traits, niche adaptation, and ecological significance of *B. equifaecis*. While maintaining our emphasis on *B. equifaecis*, we acknowledge the importance of showcasing niche adaptation as a secondary objective. We also appreciate the reviewer's suggestion to leverage comparative data. However, for a comprehensive exploration of niche differentiation, it is necessary to consider the relevant interactions of *B. equifaecis* with other microorganisms within a specific ecological niche. Therefore, our revised manuscript will present a comprehensive view of niche adaptation within a broader ecological framework. We sincerely appreciate the reviewer's valuable feedback, which has provided clear guidance for improving our manuscript.

Other comments:

Title: why not end the title with *Brachybacterium equifaecis* sp. nov.?

Re) We agree with the reviewer's suggestion and revised the title as "*Kinetic, Genomic, and Physiological Analysis Reveals Diversity in the Ecological Adaptation, and Metabolic Potential of Brachybacterium equifaecis* sp. nov. Isolated from Horse Feces"

63, rephrase - They are rarely isolated from humans, but a recent case report documented...

Re) Sentence has been rephrased as suggested.

66, rephrase - Isolation of *Brachybacterium* strains...

Re) Rephrased.

69, delete 'using...genera.'

Re) We have deleted the sentence. Thanks!

69-71, rephrase - it reads as if the authors are talking about *Brachybacterium*, but the citations are about other bacteria. If *Brachybacterium* is heterotrophic, state that first before using other heterotrophic bacteria as examples.

Re) We appreciate the reviewer's observation regarding the confusion in the cited references (69-71) and their relevance to *Brachybacterium*. To rectify this, we rephrased the sentence as follows: "*Brachybacterium is a heterotroph that must cope with natural fluctuations, such as the limited availability of nutrients and oxygen, to thrive in various environments similar to other heterotrophic bacteria (10).*" **(Lines 88-90)**

73-74, delete 'however' and make the sentence concise. Try - features contributing to niche differentiation remain to be identified.

Re) Sentence has been revised as per reviewers' suggestion.

86, Rephrase - characterizing one strain does not build a strong case for niche differentiation of a genus. However, the authors did experiments on more than one strain - take credit for that.

Re) We appreciate the reviewer's keen observation. Thus, we rephrased the sentence as “*Consequently, this study aimed to determine the physiological, kinetic, phylogenetic, and genomic properties of Brachy bacterium species, particularly emphasizing those of a novel B. equifaecis JHP9 strain isolated from horse feces.*” (Lines 104-106)

89, state the N - how many strains are used here

Re) Thank you for bringing this to our attention. In the revised manuscript, we have specified the exact number of strains to provide a clear understanding of the sample size.

90-96, this is not introduction and belongs to results and conclusions instead.

Re) We appreciate the reviewer's feedback regarding the content placement from lines 90-96. Upon reflection, we agree that this section is more appropriate for the Results and Conclusions section rather than the Introduction. In the revised manuscript, we have rephrased the last paragraph of the Introduction section to ensure better organization and flow of the paper.

104, what's the gas phase for the cultivation?

Re) Thank you for bringing this to our attention. We have included gas phase information in the updated manuscript.

111, microscopically

Re) Corrected.

121 and 124, spell out the primer sequences and cite the relevant papers.

Re) Thank you for bringing this to our attention. We have provide the primer information in the revised MS.

134, has the authors' strain been deposited into culture collection centers? If so, please provide proof.

Re) Our strain has indeed been deposited into KCTC (Korean Collection for Type Cultures), and JCM (Japan Collection of Microorganisms) as ‘KCTC 49746’ and ‘JCM 35094’ respectively. We have provided the proof of deposit in the form of a deposition certificate from KCTC and JCM as supplementary material in the revised manuscript for your reference.

166, why were the API 50 tests not carried out anaerobically?

Re) We thank the reviewer for this point. We did the API 50 CHL test under anaerobic conditions and added the information in the revised MS.

285-310, this section is a good description of JHP9, but not obvious what sets it apart from other strains.

Re) Thank you for the comment. The *Brachy bacterium* strains used in this study displayed similarities to some extent in terms of their growth conditions, nutrient utilization, enzymatic activities, and other physiological processes, indicating a conserved genetic repertoire among them. However, strain JHP9 exhibited unique traits not observed in the different strains. Specifically, JHP9 demonstrated chloramphenicol resistance (Table S4) and unique enzyme activity for alkaline phosphatase and β -glucuronidase. And also, strain JHP9 has fermentation activity of N-acetyl-D-glucosamine uniquely. These distinctive attributes of strain JHP9 likely play a significant role in its adaptation to its specific environment. We have incorporated a description of these unique features in the relevant sections of the Morphological, Physiological, and Biochemical Characterization.

293, hydrolyzed DNA? A DNase test is not mentioned anywhere in the text.

Re) We thank the reviewer for this point. We have added the method in Materials and Method section. (L 576-578)

335, the manuscript will flow better if this section goes first above all other results.

Re) Thank you for your suggestion regarding the placement of the section in question. In response to your suggestion, the revised version now incorporates the suggested change, with the section you mentioned placed at the beginning of the results.

365-368, physiology data should be taken into account as well.

Re) We appreciate your keen observation and guidance in highlighting the importance of considering physiological aspects. We have incorporated this aspect. Thanks!

373-376, what about the other *Brachy bacterium* genomes? Does this support or refute your and others' experimental results?

Re) Thank you for bringing up an important point about the prevalence of antibiotic resistance genes in *Brachy bacterium* genomes. After carefully annotating the bacterial genomes using AMRFinder, we predicted two known antibiotic resistance determinants in the JHP9 genome, specifically conferring resistance to beta-lactam and quinolone antibiotics. However, the remaining six *Brachy bacterium* genomes did not exhibit such genes. Interestingly, all strains showed resistance to kanamycin (aminoglycoside), but only JHP9 displayed resistance to chloramphenicol (phenicol) as well. The divergence in genotypic determination of antibiotic resistance genes may be attributed to the genetic context, including the presence of other genes or mutations in the genes associated with resistance to antibiotics like kanamycin and chloramphenicol. However, it is important to note that our study did not specifically identify novel genes or mutations responsible for the observed phenotypic resistance, as it was beyond

the scope of our research. Consequently, this finding contradicts the experimental results of susceptibility tests in *Brachybacterium* genomes. Nonetheless, in the revised manuscript, we have included information about the prevalence of antibiotic resistance genes across all seven *Brachybacterium* genomes. The revised passage now states:

" The JHP9 genome was found to carry three antibiotic resistance genes (ARGs) that confer resistance to two distinct groups of antibiotics. These ARGs were gyrA (Locus No.; Bequi_11645) and gyrB (Bequi_11640), encoding resistance against quinolones, while blaIII (Bequi_03615) gene determining resistance against beta-lactam antibiotics. In contrast, none of the other Brachybacterium genomes carried any of the known resistance determinants." (Lines 220-224)

388-391, Does this support or refute your and others' experimental results?

Re) We appreciate the reviewer's question regarding the support or refutation of our and others' experimental results based on the CAZymes analysis. Although we have not experimentally validated the specific functions of the carbohydrate-active genes in the *Brachybacterium* genomes, we conducted fermentation tests using six different sugars: glucose, sucrose, galactose, arabinose, lactose, and mannose. Additionally, upon determining the metabolic capability of *Brachybacterium* strains we observed that they utilize 5 different carbon sources, and ferment 20 distinctive carbohydrates (**Table S1**). It is important to note that the role of CAZymes in sugar fermentation and carbohydrate utilization can vary depending on the bacterium and its metabolic capabilities.

In the context of sugar fermentation and carbohydrate utilization, glycosyl transferases (GTs) are involved in synthesizing specific sugar-related compounds and breaking down complex carbohydrates into simpler forms that the bacterium can metabolize (1). Glycoside hydrolases (GHs) facilitate fermentation by cleaving glycosidic bonds in complex carbohydrates, releasing fermentable sugars that can be further metabolized (2). Carbohydrate esterases (CEs) may also contribute to sugar fermentation by cleaving ester bonds in certain carbohydrate structures, releasing fermentable sugars (3).

While the specific functions of these CAZymes in *Brachybacterium* species require further experimental investigation, their presence suggests their potential involvement in sugar fermentation processes. Further studies are needed to explore the precise roles and mechanisms of these CAZymes in *Brachybacterium*'s carbohydrate metabolism. We have incorporated this information in Discussion section of revised manuscript as follows:

"Although we did not experimentally validate specific functions of the predicted CAZymes, their ability to produce lactic acid from five different sugars (Figure 4), assimilate five different carbon sources, and ferment 11 distinctive carbohydrates (Table S2) implies their potential involvement in carbohydrate utilization. Nevertheless, it is crucial to acknowledge that the roles of CAZymes in sugar fermentation can differ depending on the bacterium and its metabolic capabilities." (Lines 314-319)

410, delete extra space before '!

Re) Extra space has been removed.

409-421, this paragraph should go first for this section. Then, go into the other paragraphs to present the examples of 1) shared core functions – would these support a heterotrophic and versatile lifestyle? 2) accessory functions – would this support specific niche differentiation? 3) unique functions – would this support niche adaptation?

Re) We appreciate your suggestion to restructure the paragraph order to enhance the flow of the information. In response to your recommendation, we have revised the manuscript accordingly. The paragraph you mentioned, which discusses the overall functions of *Brachybacterium* genomes, is now positioned as the introductory paragraph for this section. (L 188-203)

Furthermore, we have identified the core, accessory, and unique genomes of *Brachybacterium* contributing to the presence of complete or nearly complete metabolic pathways. These findings have been discussed in the revised manuscript, highlighting the potential roles of these genomes in facilitating a versatile lifestyle, niche differentiation, and niche adaptation. **(Lines 192-203)**

423, how does this relate to the authors' intro about K vs S strategies?

Re) We thank the reviewer for this point. Actually, *K-strategist* microbes grow slowly but can become competitive under low substrate concentration due to their high substrate affinities. We have added the point in Discussion section of revised manuscript as follows:

“Our results imply that Brachybacterium, including strain JHP9, have a high affinity for glucose and oxygen. The K-strategist microbes grow slowly but can compete at low substrate concentrations because of their high substrate affinity (49). Therefore, members of the family Dermabacteriaceae, including strain JHP9, tend to utilize glucose and oxygen to survive in temperate ecosystems; hence, such a high substrate affinity is advantageous to allow them to adapt and colonize the host while competing with specialists in the same niche (57). Furthermore, the high GC content observed in Brachybacterium genomes (Table 1) contributes to the stability of their DNA (40). This characteristic implies that they have the capacity to flourish within specific temperature ranges and adapt to their ecological niches.” **(Lines 431-439)**

424-437, citations when referring to others' work.

Re) Thank you for the comment. We have cited proper references.

439, this section should discuss first the discovery and characterization of a novel species. It should also be rewritten depending on the author's priorities - niche adaptation or niche differentiation.

Re) We appreciate reviewers' suggestion and have rewritten this section as follows:

“In this study, we described the phenotypic and genotypic characteristics of a novel B. equifaecis JHP9 strain, along with those of closely related Brachybacterium strains, using physiological

and genomic analyses. Interestingly, Brachy bacterium species possess properties that facilitate their versatile lifestyles and ecological adaptations. These characteristics encompass their ability to survive under various temperature, pH, and salt concentration conditions, their ability to utilize a variety of carbohydrates, and their high affinity for oxygen and glucose. Collectively, these traits facilitate successful adaptation and survival in specific environmental settings. Furthermore, analysis of the novel JHP9 strain revealed unique factors contributing to its defense mechanism, including antibiotic resistance, relatively high affinity for oxygen and glucose, and efficient utilization of sugars. These traits reflect the adaptation of strain JHP9 to the niche environment of the horse intestine.” (Lines 290-300)

442-444, it's not obvious from the results what set JHP9 apart from other strains, and what is considered 'broad range'.

Re) Regarding the reviewer's comment, we acknowledge the need for clarification regarding distinctive features of the strain JHP9 and what is meant by a "broad range". We have revised the paragraph to focus on the comparative analysis of the *Brachy bacterium* genomes, as well as incorporated distinctive features of the novel strain. Moreover, the term "broad growth range" refers to the ability of the organisms to thrive and function across a wide spectrum of temperature, pH, and salt concentration conditions. Upon careful evaluation, we recognized that although *Brachy bacterium* exhibits tolerance to a range of temperature, pH, and salt concentrations, its optimal growth occurs within a relatively narrow range for each parameter. As a result, we have replaced the term "broad range" to reflect this observation. Please refer to our previous response for further clarification.

446, has it been detected in other environments beside horse feces? An alternative explanation is these features (particularly when not shared with other *Brachy bacterium*) are there for niche adaptation in the horse intestine.

Re) We thank the reviewer for the suggestion. This is the first report on the isolation of strain JHP9 from horse feces, and it has not been identified from any other source yet. In addition, we have not done the sequence recruitment of the *Brachy bacterium* sequence in the database from the horse feces either. However, different *Brachy bacterium* strains with similar growth features have been identified from various environments. Thus, we assume this novel strain may also survive in other environments. However, as suggested by the reviewer, it is more appropriate to write about its niche adaptation in the horse intestine, keeping in view its distinctive features of defense (phenotypic resistance to chloramphenicol and resistance determinants for quinolone), relatively high kinetic affinity, and efficient utilization of sugars. Thus, we modified the sentence as:

“Furthermore, analysis of the novel JHP9 strain revealed unique factors contributing to its defense mechanism, including antibiotic resistance, relatively high affinity for oxygen and

glucose, and efficient utilization of sugars. These traits reflect the adaptation of strain JHP9 to the niche environment of the horse intestine.” (Lines 296-300)

447, delete 'technically'

Re) We have deleted the “technically”.

451, physiological

Re) The typo has been rectified. Thanks!

470, what are the inhabiting environments?

Re) Here, by inhabiting the environment, we intended a living environment or habitat for the isolates. We have modified the paragraph to include the involvement of core, accessory, and unique genomes in the necessary pathways and their implications for bacterial ecological adaptation. Please check the revised section.

495, conversely

Re) We replaced reversely with conversely.

496, The genes may well be expressed but perhaps, there is a dose effect undetected by the authors' assays. Clinical breakpoints are species-specific, the MIC for the same antibiotic can vary among pathogens. It's also not black and white - between resistant and susceptible, there is also intermediate. Were the horses being treated regularly with antibiotics and what kinds?

Re) We agree with the reviewer that alteration in the antibiotic dose (potency) may alter the phenotypic expression of the genes. However, this study aimed at determining the susceptibility of the *Brachybacterium* isolates against the FDA (Federal Drug Agency) approved potency of antibiotics which are commonly used in animals for therapeutic or prophylactic purposes (<https://www.fda.gov/animal-veterinary/products/approved-animal-drug-products-green-book>). Here, we applied commonly used doses of antibiotics ampicillin (10 µg), chloramphenicol (30 µg), gentamicin (10 µg), kanamycin (30 µg), streptomycin (10 µg), and tetracycline (30 µg). We also agree with the reviewer that MIC values may vary among species. We interpreted the zones of inhibition based on the CLSI-prescribed values of MIC for susceptible, intermediate, and resistant isolates. According to our results, MIC values for all the tested *Brachybacterium* isolates either fall in the range of tolerant or resistant. There were no intermediates. Furthermore, the lack of phenotypic resistance associated with these genes may be attributed to factors such as mutations or genetic variations within the genes or their regulatory elements, the absence of specific inducers or repressors, and genetic regulatory mechanisms (ref). We reflected these changes on the selection of antibiotics and non-expression of the genes in the material and methods, and discussion, respectively, as follows:

“These antibiotics were selected on the basis of their common use for therapeutic and prophylactic purposes in horse (<https://www.woah.org/app/uploads/2021/03/oie-list-antimicrobials.pdf>).” (Lines 598-600)

“The lack of phenotypic resistance associated with the beta-lactam-resistance gene, blaIII, may be attributed to several factors. These factors include mutations or genetic variations within the gene or its regulatory elements, absence of specific inducers or repressors, and genetic regulatory mechanisms (38-40).” (Lines 367-370)

582, potential utilization... unless it's already been used in food industry.

Re) Thank you for the comment. It has not already been utilized in the food industry. Hence, we agreed with the reviewer and revised the sentence by incorporating ‘potential utilization’.

Table 1 - any comments on the high GC content of *Brachybacterium*?

Re) We appreciate the reviewer for posing this insightful question. The higher GC content contributes to increased stability of the DNA molecule, making it more resistant to heat denaturation. This enables the bacteria to thrive in their specific temperature range and adapt to their niche (ref). We have incorporated this point in the manuscript as follows:

*“Furthermore, the high GC content observed in *Brachybacterium* genomes (Table 1) contributes to the stability of their DNA (40). This characteristic implies that they have the capacity to flourish within specific temperature ranges and adapt to their ecological niches.” (Lines 436-439)*

Figure 1A - highlight the authors' strain in bold, and underline the species selected for comparative analysis. Figure legends have A and B mixed up.

Re) We appreciate the reviewer's comment regarding Figure 1A. We apologize for the mix-up in the figure legends, and we have rectified the issue by appropriately repositioning the figure legends. In the revised figure, we have highlighted the strain JHP9 in bold and underlined the species chosen for comparative analysis.

Figure 2A - color keys?

Re) In Figure 2A, the labels are indicated next to the corresponding ring colors, as well as in the matrix. However, for enhanced clarity, we have made incorporated this information in the figure legends as follows:

“The deep ring colors represent the presence of the respective gene clusters, whereas faded ring colors indicate their absence.”

Figure 2B - unit for Y axis?

Re). We have included the unit for the Y-axis in Figure 2B.

Figure 3B - why the big variation for *E. coli*?

Re) We thank the reviewer for this point. The values were from different studies with different strains of *E. coli*, which were tested in various experimental conditions; therefore, it could make this like big variations. We cited the reference study for *E. coli* in the revised MS, so please check this.

Figure 4 - nice figure but a summary of the main take aways should be included in the figure legends.

Re) We appreciate the reviewer's feedback on the figure. In response to this suggestion, we have revised the figure legends to include a summary of the key findings and main implications depicted in the figure.

“This figure illustrates the genomic potential of strain JHP9 in the uptake of diverse carbohydrates, minerals, and metal ions. It also highlights the involvement of specific proteins in the ABC transportation system for oligosaccharides, monosaccharides, minerals, phosphates, and metallic cations. Additionally, the figure showcases the enzymes associated with the utilization of various carbohydrates, such as galactose, glucose, trehalose, glycogen, lactose, and cellobiose metabolism, as well as pathways such as the pentose phosphate pathway, pyruvate metabolism, amino acid metabolism (arginine, ornithine, proline, and asparagine), cofactors and vitamins (CoA and heme biosynthesis), terpenoid biosynthesis, glycerolipid metabolism, and oxidative phosphorylation.”

Suppl. Fig. S5 - the oxygen data is excellent, but the glucose data is patchy. In particular, this reviewer does not feel confident about the glucose data for *B. nesterenkovi* and *B. huguangmaarens*.

Re) Thank you for the point, and we also agree with that point. As follow reviewer's comment, we did more experiments with the strains, and the updated results are included in the revised manuscript.

References

1. Lairson LL, Henrissat B, Davies GJ, Withers SG. 2008. Glycosyltransferases: structures, functions, and mechanisms. *Annu Rev Biochem* 77:521-55.
2. Pengthaisong S, Piniello B, Davies GJ, Rovira C, Ketudat Cairns JR. 2023. Reaction Mechanism of Glycoside Hydrolase Family 116 Utilizes Perpendicular Protonation. *ACS Catalysis* 13:5850-5863.
3. Christov LP, Prior BA. 1993. Esterases of xylan-degrading microorganisms: Production, properties, and significance. *Enzyme and Microbial Technology* 15:460-475.

July 31, 2023

Prof. Man-Young Jung
Jeju National University
Department of Science Education
102 Jejudaehak-ro
Jeju-si, Jeju-do 63243
Korea (South), Republic of

Re: Spectrum05048-22R1 (Kinetic, Genomic, and Physiological Analysis Reveals Diversity in the Ecological Adaptation and Metabolic Potential of *Brachybacterium equifaecis* sp. nov. Isolated from Horse Feces)

Dear Prof. Man-Young Jung:

Your manuscript has been accepted, and I am forwarding it to the ASM Journals Department for publication. You will be notified when your proofs are ready to be viewed.

Sincerely,

Jeffrey Gralnick
Senior Editor, Microbiology Spectrum
